# Single-molecule characterization of subtype-specific β1 integrin mechanics

Myung Hyun Jo [1,11], Jing Li[2,3,11], Valentin Jaumouillé [4,9], Yuxin Hao[2,3], Jessica Coppola[5,10], Jiabin Yan[2,3], Clare M. Waterman [4], Timothy A. Springer [2,3,12] & Taekjip Ha [1,6,7,8,12]

Although integrins are known to be mechanosensitive and to possess many subtypes that have distinct physiological roles, single molecule studies of force exertion have thus far been limited to RGD-binding integrins. Here, we show that integrin α4β1 and RGD-binding integrins (αVβ1 and α5β1) require markedly different tension thresholds to support cell spreading. Furthermore, actin assembled downstream of α4β1 forms cross-linked networks in circularly spread cells, is in rapid retrograde flow, and exerts low forces from actin polymerization. In contrast, actin assembled downstream of αVβ1 forms stress fibers linking focal adhesions in elongated cells, is in slow retrograde flow, and matures to exert high forces (>54-pN) via myosin II. Conformational activation of both integrins occurs below 12-pN, suggesting that post-activation subtype-specific cytoskeletal remodeling imposes the higher threshold for spreading on RGD substrates. Multiple layers of single integrin mechanics for activation, mechanotransduction and cytoskeleton remodeling revealed here may underlie subtype-dependence of diverse processes such as somite formation and durotaxis.

Integrins integrate the intracellular and extracellular environments to mediate cell adhesion, migration, and formation of tissues. Integrins have three conformational states. For unliganded integrins on the cell surface, the bent-closed (BC) state greatly predominates over the extended-closed (EC) and extended-open (EO) states. Integrin extracellular domains bind to ligands on the surface of other cells or in extracellular matrix, while their cytoplasmic domains bind to cytoskeletal-associated adaptor proteins. When force, exerted by the cytoskeleton on the integrin β-subunit cytoplasmic domain, is resisted by a ligand embedded in the extracellular environment, tensile force is applied through the integrin and stabilizes the extended, EC and EO states over the bent, BC state. Ligand binding also stabilizes the high affinity, open, EO state over the low affinity, closed, BC and EC states[1–7]. Furthermore, tensile force stabilizes association of cytoskeletal components and signaling proteins into multi-layered, complex assemblies. Consequently, integrin-mediated adhesions grow and mature in response to cytoskeletal forces or increased substrate rigidity, and shrink or disassemble in the absence of such stimuli[8].

Molecular tension sensors, with fluorescent reporter systems, such as peptide-based tension sensor[9], DNA hairpin-based digital tension sensor[10] and the DNA double helix-based rupturable tension gauge tethers (TGTs)[11] have been developed to measure the force

[1]Department of Biophysics and Biophysical Chemistry, Johns Hopkins University, Baltimore, MD 21205, USA. [2]Program in Cellular and Molecular Medicine, Boston Children's Hospital, Boston, MA 02115, USA. [3]Department of Biological Chemistry and Molecular Pharmacology, Harvard Medical School, Boston, MA 02115, USA. [4]Cell and Developmental Biology Center, National Heart Lung and Blood Institute, National Institutes of Health, Bethesda, MD 20814, USA. [5]Institute for Protein Innovation Harvard Institutes of Medicine, Boston, MA 02115, USA. [6]Department of Biophysics, Johns Hopkins University, Baltimore, MD 21205, USA. [7]Department of Biomedical Engineering, Johns Hopkins University, Baltimore, MD 21205, USA. [8]Howard Hughes Medical Institute, Baltimore, MD 21205, USA. [9]Present address: Department of Molecular biology and Biochemistry, Simon Fraser University, Burnaby, BC V5A1S6, Canada. [10]Present address: Department of Immunology and Microbiology, The Scripps Research Institute, La Jolla, CA 92037, USA. [11]These authors contributed equally: Myung Hyun Jo, Jing Li. [12]These authors jointly supervised this work: Timothy A. Springer, Taekjip Ha. ✉e-mail: springer@crystal.harvard.edu; tjha@jhu.edu

exerted by a single integrin on a ligand. The TGT exploits the physical rupture force of double-stranded DNA to measure or limit the force applied through single receptor-ligand bonds over a wide range of tension (12–56 pN). DNA duplex of TGT withstands up to a certain level of tension and undergoes irreversible rupture above the tension tolerance. Thus, the peak force through a single integrin, rather than instantaneous force, is controlled and measured. The TGT conjugated with cRGDfK peptide revealed that resistance to forces over ~40 pN is required for cell spreading through RGD-binding integrins on multiple cell lines[11]; subsequent studies of other cell types also observed the same tension threshold[12,13]. However, the large distance change and energetics associated with integrin activation suggested that constant force as low as 2 pN applied across a single integrin-ligand bond could be enough to induce a conformational change to the EO state[14]. Therefore, the molecular basis for what appears to be the common tension threshold for initiating adhesion and spreading responses through single RGD-binding integrin-ligand bonds remains unclear.

RGD-binding and laminin-binding integrins are usually involved in cell-matrix adhesion and many of them form focal contacts; all previous molecular tension sensors examined only the RGD-binding integrins. Other integrin classes have also been implicated in mechanotransduction. For example, LFA-1 (αLβ2) and VLA-4 (α4β1) provide the traction for migration of lymphocytes and other leukocytes which move rapidly compared to fibroblasts. Lymphocyte actin does not form thick actomyosin bundles or associate with focal contacts as seen in fibroblasts[15]. The average force on LFA-1 integrins in lymphocytes was reported to be low, for example, ~1.5 pN, although this measurement was on an ensemble of integrin molecules and may have been due to a small fraction of integrins engaged with ligands that experience much higher forces[3]. However, low time-averaged forces of 1–3 pN have also been reported on RGD substrates[9]. It is currently unknown whether different integrins and the cytoskeletons they engage exert different magnitudes of tensile force and respond to substrate rigidity differently. It is also unknown whether the tension requirement for cell spreading is set by the force required to induce conformational activation of the integrin to the EO state or by the force required to stabilize cytoskeleton assembly.

Here, to test if the mechanical force required to signal through integrins is subtype-specific, we developed novel TGTs conjugated with MUPA-LDVPAAK peptide[16], an α4β1-specific peptidomimetic ligand (LDVP-TGT). We compared cellular responses to LDVP-TGT and TGTs conjugated with cRGDfK peptide (RGD-TGT), an established benchmark[11], using the foreskin fibroblast cell line, BJ-5ta, which natively expresses integrin α4β1 and four different RGD-binding integrins. α4β1 is co-expressed with α5β1 and αV integrins both on immune cells and a wide range of non-hematopoietic cell types including fibroblastic, endothelial, melanoma, and rhabdomyosarcoma cells[17]. Integrin α4β1 binds to an alternatively spliced region in fibronectin as well as to the cell surface ligand, vascular cell adhesion molecule (VCAM).

We find that the cytoskeleton assembled by integrin α4β1 requires a tension of less than 12 pN to activate cell spreading, distinct from the cell type-independent threshold of ~40 pN for RGD-binding integrins[11]. Furthermore, BJ-5ta cell adhesion sites anchored by RGD-binding integrins, mainly through αVβ1, an underexplored integrin subtype in the field, require and exert much higher peak forces than those anchored by integrin α4β1. The force required for cell spreading mediated by αVβ1 is also much higher than the force required for its conformational activation. These conclusions from single-molecule level TGT rupture measurements are further supported by the higher bulk forces exerted by RGD-binding integrins on substrates than by integrin α4β1. We also detect a marked difference in cell morphology, cytoskeleton architecture, actin retrograde flow speed and adhesion site distribution for BJ-5ta cells adhering through integrin α4β1 compared to αVβ1. In addition, from time-resolved molecular tension

measurement using quenched TGT (qTGT)[18], we trace complete single molecule nanomechanical histories from the moment of cell landing on the substrate to the final stage of cell spreading. Lastly, we conjugated TGTs with a distinct RGD peptide with higher affinity for α5β1, cyclic-ACRGDGWCGK, and showed that α5β1, like αVβ1, is sufficient to mediate cell spreading, elongation, and attain high tension. Our results demonstrate interplay between chemical sensing and mechanical sensing of cellular environments through different integrin subtypes, which determine the assembly of cytoskeletons with distinctive architectures and tensile force exertion on the substrate.

## Results

### Cells spread and migrate differently on RGD- and LDVP-TGT

To compare cell adhesion and force exertion mediated by different integrin subtypes, we covalently conjugated subtype-specific peptidomimetic ligands to one strand of a short double-stranded DNA TGT[11,19], with the other strand modified with biotin at distinct positions to withstand distinct rupture tension thresholds (Fig. 1a, Supplementary Fig. 1a; see Methods for estimation of tension tolerance). For RGD-binding integrins, we used cyclo[Arg-Gly-Asp-D-Phe-Lys] (cRGDfK), a cilengitide analog, which has high affinity for αV integrins[20]. For integrin α4β1, we used a well-validated peptidomimetic ligand specific for α4β1[16,21] that contains a 2-methylphenylureaphenylacetyl (MUPA) moiety linked to the Leu-Asp-Val-Pro (LDVP) motif from the fibronectin III CS segment (MUPA-LDVPAAK) and binds to the EO state of α4β1 with high affinity (0.15 nM)[22].

We first immobilized RGD-TGT or LDVP-TGT with 54 pN tension tolerance (RGD-54pN or LDVP-54pN) on PEG-passivated surfaces at a density of ~1500 μm⁻² and examined adhesion, spreading and migration of BJ-5ta[23], a hTERT-immortalized human foreskin fibroblast cell line that natively expresses both α4β1 and RGD-binding integrins (Supplementary Fig. 1b, c). Cells were viewed with differential interference contrast (DIC) microscopy (Supplementary Fig. 1d), with confocal microscopy (Supplementary Fig. 2) and with reflection interference contrast microscopy (RICM) to monitor the topography of the ventral cell membrane. While cells started to adhere and spread on both surfaces within a few minutes of landing (Movie 1), by one hour, their morphologies differed markedly (Fig. 1b). On RGD-54pN, cells developed elongated, irregular shapes. In contrast, on LDVP-54pN, cells spread uniformly in simple circular shapes.

We confirmed the ligand and integrin specificity of cell adhesion by using soluble cRGDfK or MUPA-LDVPAAK peptide as a competitor (Fig. 1c, Supplementary Fig. 3a). Cell spreading was specifically inhibited by the cognate ligand but not influenced by the dissimilar ligand. Cell spreading was clearly mediated by ligand-TGTs because cells spread on LDVP-54pN surface promptly detached when the DNA tether was digested with DNase I (Supplementary Fig. 3b; Movie 2).

Distinctive cell morphologies on RGD- and LDVP-TGT were mimicked by physiologic, macromolecular ligands (Fig. 1b, d). On ligands with RGD motifs, including vitronectin, proTGF-β1, and plasma fibronectin, cells were elongated with irregular lamellipodia. On the integrin α4β1 ligand, VCAM, BJ-5ta cells spread into uniform, round shapes. The cognate macromolecular ligands thus recapitulated the BJ-5ta cell spreading phenotypes on RGD and LVDP-TGTs, as quantitatively confirmed by spreading aspect ratios (Fig. 1e). This observation suggests that the distinctive cell morphology we observed on LDVP-TGT compared to RGD-TGT is not due to the affinity difference between different integrin and ligand pairs, as VCAM binds to α4β1 with ~300 fold lower binding affinity compared to the peptidomimetic ligand, LDVP[22], yet the two substrates support the same cell spreading morphology.

In another contrast, BJ-5ta cells migrated on RGD-54pN but not on LDVP-54pN. Cells on RGD-54pN became protrusive over time and started to migrate, with morphologically distinct leading and trailing edges (Fig. 1f). In contrast, cells spreading on LDVP-54pN maintained

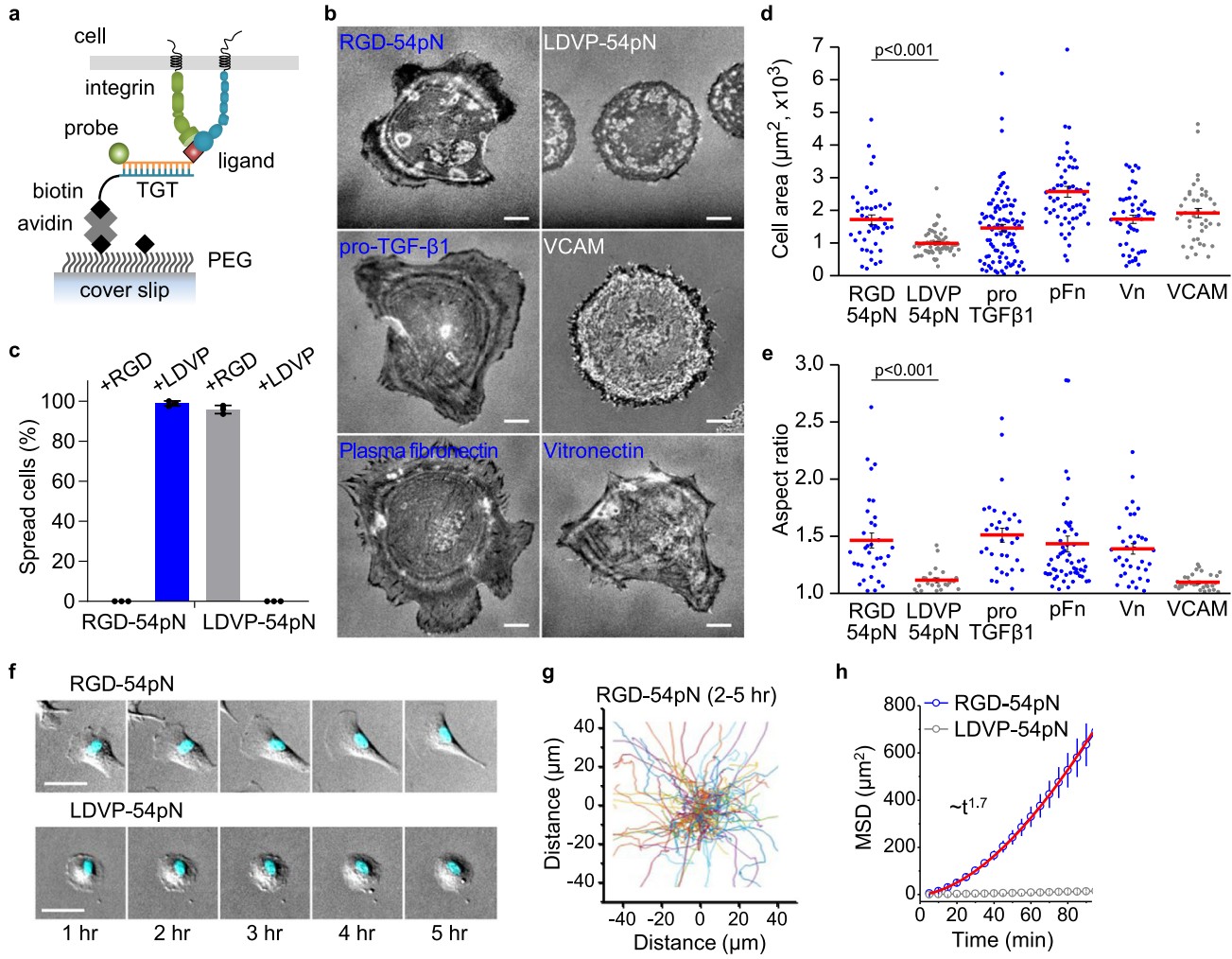

**Fig. 1 | Ligand dependent cell spreading and locomotion. a** Schematic of TGT presenting a ligand for integrins. TGT-54pN (Cy3) immobilized on PEG-passivated surface is depicted. **b** Representative reflection interference contrast microscopy (RICM) images of cells seeded (1 h) on surfaces with immobilized TGT or adsorbed biological ligands. Scale, 10 μm. **c** Percentage of spread cells on LDVP-54pN or RGD-54pN surfaces (mean ± SD; $n = 3$ independent experiments). Cells were seeded with soluble ligands (LDVP or RGD peptides, 100 μM) in solution. **d** Close-contact enclosed areas from RICM images (mean ± SE; $n = 43, 64, 96, 59, 50, 41$ cells). **e** Aspect ratio of the enclosed areas (mean ± SE; $n = 33, 24, 32, 56, 36, 33$ cells,

>500 μm²). **f** Cell morphology change and locomotion over time. Hoechst signals (cyan) overlaid on differential interference contrast (DIC) images (gray) show nuclei. Scale, 50 μm. **g** Cell nucleus trajectories (2–5 h after seeding) on RGD-54pN surface. **h** Mean square displacement ($n = 114$ (RGD-54pN) and 164 (LDVP-54pN) cells, mean ± SD). The red line shows the fitted power-law curve. Two-sided $t$-test for $p$-values. All results are representative of multiple experiments. Data are combined from three independent experiments. Source data are provided as a Source Data file.

the same circular shape and same position for more than five hours. To quantify cell movements, cell nuclei stained with Hoechst were tracked (Fig. 1g). The mean squared displacement increased with time t according to ~t[1.7] on RGD-54pN (Fig. 1h), whereas cells on LDVP-54pN did not migrate.

**Tension threshold for cell spreading on RGD- and LDVP-TGT**
To define the molecular tension that single integrin-ligand bonds need to withstand to initiate cell spreading, we used TGTs with different tension tolerance, $T_{tol}$, values (Fig. 2a). Cells spread well on RGD-43pN and RGD-54pN, but remained spherical on 12, 23 and 33 pN RGD-TGT (Fig. 2b, c; Movie 3). Thus, RGD-mediated cell spreading requires tensions >33 pN, consistent with the previous study of RGD-TGT on other adherent cell lines[11–13]. In contrast, cells adhered and spread well on LDVP-TGT at all $T_{tol}$ values and formed the characteristic circular shape, although spread area increased at 43 and 54 pN $T_{tol}$ values (Fig. 2d). Overall, these data show that for BJ-5ta cells, α4β1-mediated adhesion and spreading requires a much lower tension threshold (<12 pN) than that through RGD-binding integrins (33–43 pN).

Fibronectin contains an RGD-integrin binding site and also an integrin α4β1 binding site with an LDVP motif[24]. We mimicked interaction between these sites by studying adhesion to substrates bearing both types of TGTs. Cells plated on the LDVP-54pN surface for one hour spread and formed the characteristic circular shape. Subsequently added RGD-TGT (Fig. 2e) was promptly immobilized on the surface, except for the region masked by spread cells (Supplementary Fig. 3c), and triggered further cell spreading and switching to the same irregular shape, characteristic of spreading on the RGD surfaces (Movie 4). This dramatic morphological change, however, was not observed when RGD-TGT bearing no biotin strand was added (Fig. 2f, 0pN), showing that ligand binding in the absence of force transmission through the integrin cannot trigger switching to the elongated morphology. Even more strikingly, the morphological switch was not observed when RGD-TGT with lower tension tolerance was added (Fig. 2f; 12, 23 and 33 pN). Thus, the mechanosensing systems for spreading on RGD- and LDVP-TGT are independent of one another, and the presence of mature integrin α4β1-based adhesions on LDVP-54pN does not

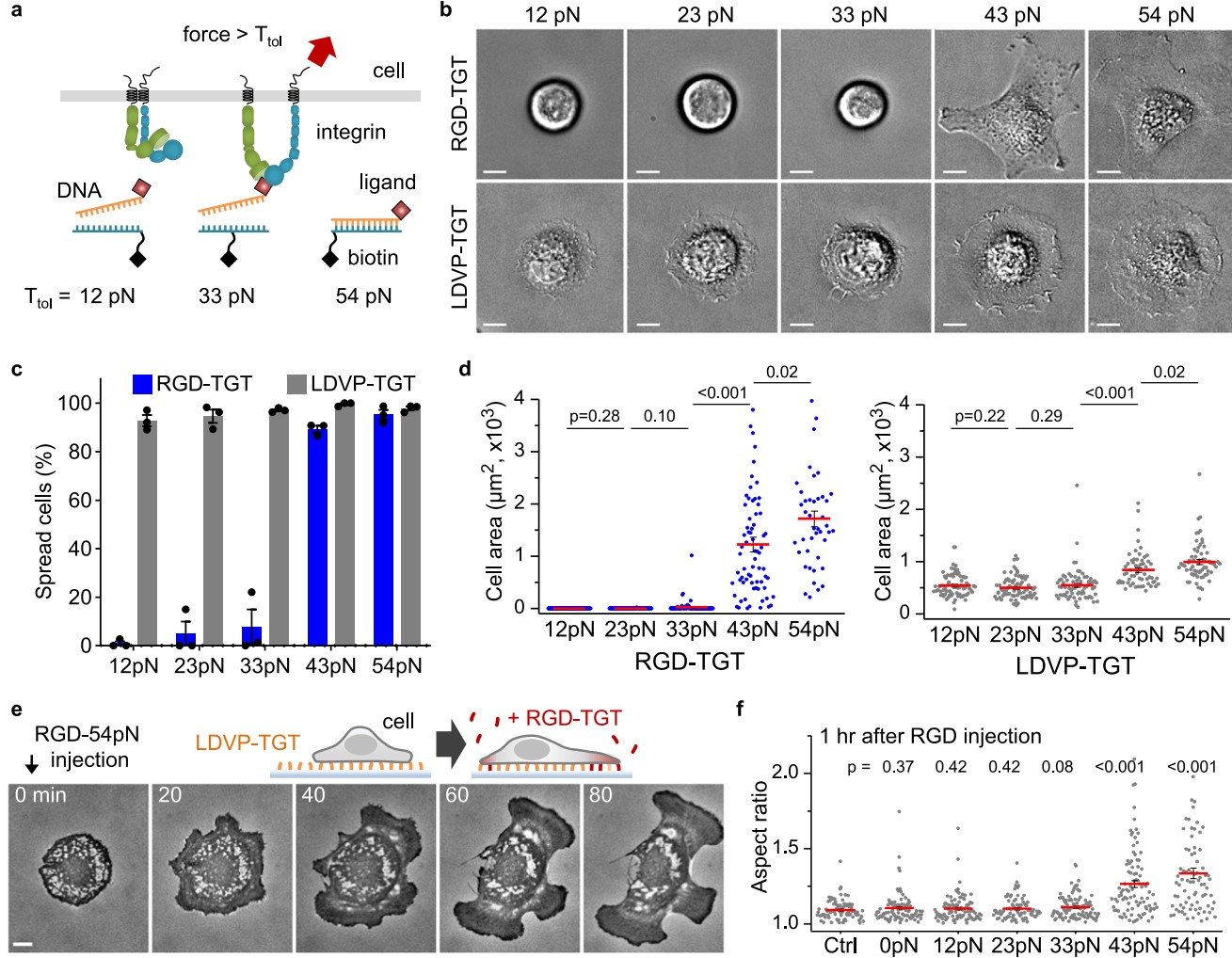

**Fig. 2 | Ligand and molecular tension-dependent cell spreading. a** Schematic depicting the rupture of TGT by integrin force. **b** Representative BJ-5ta cells on TGT surfaces 1 h after cell seeding (DIC images). Scale bars, 10 μm. **c** Percentage of spread cells on TGT (mean ± SE; $n = 3$ independent experiments). **d** Close-contact enclosed area of cells (mean ± SE; $n = 96, 82, 97, 70, 43, 75, 77, 70, 58, 64$ cells). **e** Time-lapse RICM images. RGD-54pN was added to the chamber with cells spreading on LDVP-54pN. Scale bar, 10 μm. See also Movie 4. **f** Aspect ratio of cell-enclosed areas 1 h after RGD-TGT (100 nM) injection (mean ± SE; $n = 84, 94, 97, 83, 104, 94, 80$ cells). Control (Ctrl) or 0 pN indicates the injection of cell medium or RGD conjugated TGT without biotinylated strand. Two-sided $t$-test for $p$-values. All results are representative of multiple experiments and combine measurements on cells from three independent experiments. Source data are provided as a Source Data file.

affect the tension threshold required for cytoskeleton assembly in response to RGD ligands.

## Single-molecular force exertion on RGD- and LDVP-TGT

We next estimated the level and spatial distribution of tensile forces during cell adhesion and spreading by measuring TGT rupture events. Ruptured TGTs are better detected over the background of non-ruptured TGT using quenched TGT (qTGT or turn-on TGT)[18,25]. In qTGT, fluorescence of a probe attached to the biotinylated DNA strand is unquenched when the TGT is ruptured by removal of the complementary DNA strand bearing both the quencher and the ligand (Fig. 3a). qTGT rupture was quantified using total internal reflection fluorescence microscopy (TIRFM) and calibrated by single probe intensity (Supplementary Fig. 4a). RGD-qTGT rupture showed streak patterns mostly located at the periphery of spread cells (Fig. 3a, b, Supplementary Fig. 4b), with rupture extending to somewhat higher radial distances on RGD-54pN than RGD-43pN suggesting that high tension was transmitted by maturing focal adhesions (Fig. 3b). Significantly less rupture was observed for RGD-

54pN than RGD-43pN (Fig. 3c), showing that much of the high peak force exerted by RGD-binding integrins during spreading and migration is between 43 and 54 pN. Rupture events of 12, 23 and 33 pN RGD-TGT were relatively fewer because of the much smaller cell contact area. However, there were significantly more rupture events of RGD-12pN TGT in the absence than in the presence of competing soluble cRGDfK ligand (100 μM) (Supplementary Fig. 5a). Furthermore, there was significantly more rupture of RGD-23pN than that of RGD-12pN and more rupture of RGD-33pN TGT than that of RGD-23pN (Fig. 3c). These results demonstrate peak force exertion during abortive cell spreading attempts increase in frequency as the threshold to support cell spreading is approached. Abundant RGD-12pN rupture was observed when cell spreading was facilitated by adding RGD-54pN (Supplementary Fig. 5b), indicating that less rupture of RGD-12pN is not the consequence of low ligand accessibility or geometrical difference compared to TGTs of higher tension tolerance. This finding is consistent with comparisons of alternative TGT geometries which showed that differences in cell adhesion on RGD-TGT of different tension tolerance values is not a result of

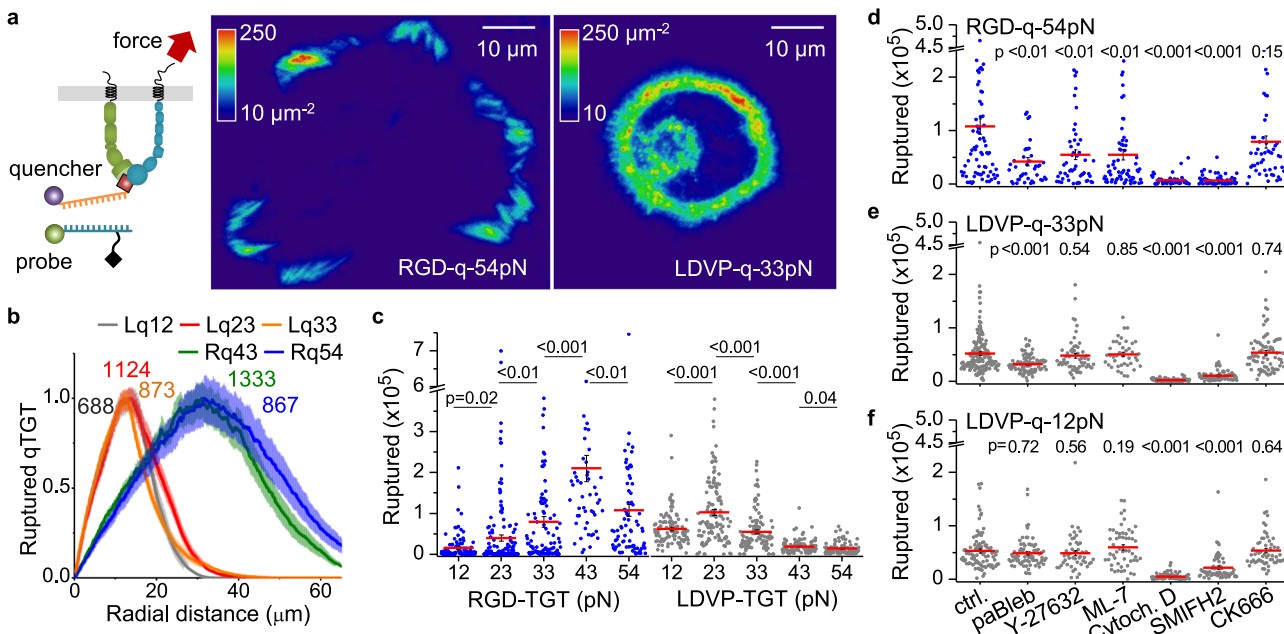

**Fig. 3 | Integrin tension-induced TGT rupture. a** TGT rupture density maps locating high force transmission (>$T_{tol}$) events cumulated for 1 h (see color key). Scale bar, 10 μm. Schematic of qTGT is depicted. The signals of ruptured TGT were calibrated using single-molecule dye intensity. **b** Normalized radial profiles of ruptured TGT counts (mean ± SE; $n$ = 104, 112, 215, 96, 112 cells). The maximum values before normalization are noted on the curves. Lq12 or Rq43 indicates LDVP-12pN or RGD-43pN (BHQ2-Cy3), respectively. **c** Rupture events per cell ($n$ = 106, 168, 104, 53, 79, 104, 118, 116, 110, 91 cells). Note that the cumulated rupture event counts on RGD-12/23/33pN surfaces, which do not allow the initiation of cell

spreading, should not be directly compared to the other conditions. **d–f** Ruptured TGT per cell with cytoskeletal inhibitors on the indicated RGD-54pN ($n$ = 79, 42, 51, 68, 48, 71, 51 cells), LDVP-33pN ($n$ = 164, 94, 58, 41, 94, 73, 76 cells), and LDVP-12pN ($n$ = 76, 49, 53, 66, 62, 57 cells). Para-amino-blebbistatin, paBleb. Cytochalasin D, Cytoch. D. All results are representative of multiple experiments and combine measurements on cells from three or more independent experiments. Lines show mean ± SE. Two-sided $t$-test for $p$-values. Source data are provided as a Source Data file.

---

differences in ligand accessibility or steric hindrance from dsDNA or neutravidin[11].

In contrast to RGD-qTGT, abundant 12, 23 and 33 pN LDVP-qTGT rupture occurred both inside and near the edge of spread cells (Fig. 3a, b). Rupture events per cell was highest at 23 pN and decreased significantly at 33 pN and further at 43 and 54 pN (Fig. 3c). Interestingly, rupture events were significantly more frequent on LDVP-23pN than on LDVP-12pN, despite similar cell spreading (Fig. 2d). This behavior shows that the cytoskeletal machinery assembled by α4β1-based adhesions adapts to higher force resistance by applying greater force on LDVP-23pN than on LDVP-12pN. However, this adaptive response was limited as it was not seen at tension tolerance values of 33–54 pN.

Cellular force transmitted through integrins is generated by actin polymerization, with or without actomyosin activity[26]. To identify the key cytoskeletal components involved in the force exertion through these integrins, we examined the effects of cytoskeletal inhibitors on the rupture of TGT (Fig. 3d–f, Supplementary Fig. 6). Cytochalasin D, which binds to the fast growing, barbed end of actin filaments to prevent their polymerization, inhibited cell spreading and eliminated TGT rupture on all substrates. Formin inhibitor SMIFH2, which inhibits formin-mediated actin nucleation and growth in linear filaments, also greatly diminished TGT rupture. Furthermore, SMIFH2 greatly decreased spreading on RGD- and LDVP-TGT. The Arp2/3 inhibitor CK666, which blocks branched actin nucleation in lamellipodia, had no significant effect. These results show that formin-mediated actin polymerization is essential for generating cytoskeletal forces during cell spreading for both integrin subtypes.

We also interfered with components of actomyosin contractility by blocking the actin-binding ATPase of the myosin-II head with para-amino-blebbistatin, blocking Rho-associated protein kinase (ROCK) with Y-27632, and blocking myosin light chain kinase (MLCK) with

ML-7. These three inhibitors did not disrupt cell spreading on any of the substrates tested (Supplementary Fig. 6), suggesting that actomyosin-generated force is not essential in early-stage cell spreading for either integrin subtype. However, the rupture of RGD-54pN was significantly suppressed by each of these three inhibitors (Fig. 3d), indicating that actomyosin contraction exerts forces above 54 pN through RGD-binding integrins, as previously shown for CHO-K1 and 3T3 cells[19,27]. Although the effect of the same inhibitors was absent or lower on LDVP-TGT, a most interesting contrast was seen with para-amino-blebbistatin, which significantly reduced rupture events on LDVP-33pN but not on LDVP-12pN (Fig. 3e, f). These results show that myosin II contributed to force generation in the actin cytoskeleton assembled downstream of α4β1 engagement by LDVP-33pN, but not by LDVP-12pN, and agree with the finding above that the α4β1-engaged cytoskeleton can sense strain on the substrate and remodel to exert higher force on a rigid substrate. Overall, these results show that for both integrin subtypes, early-stage cell spreading is driven by actin polymerization and subsequent development of high force transmission is myosin II contraction-dependent.

## Integrin-subtype dependent traction force and actin flow

We used traction force microscopy to test if the markedly different single-molecule forces exerted by the cytoskeletons engaged to RGD-binding and α4β1 integrins carried over to different cellular traction forces and actin retrograde flow speeds. On both RGD-coated 4 kPa and 0.7 kPa gels, cells showed clear centripetal traction forces in leading edge regions (Fig. 4a–c and supplementary Fig. 7a, b). Traction force was significantly lower on LDVP-coated gels and was clearly observable only on the soft gel (Fig. 4c, note the log scale). Despite the ~10-fold lower traction force, integrin α4β1 still mediated cell spreading on the soft gel with the characteristic circular shape (Fig. 4b).

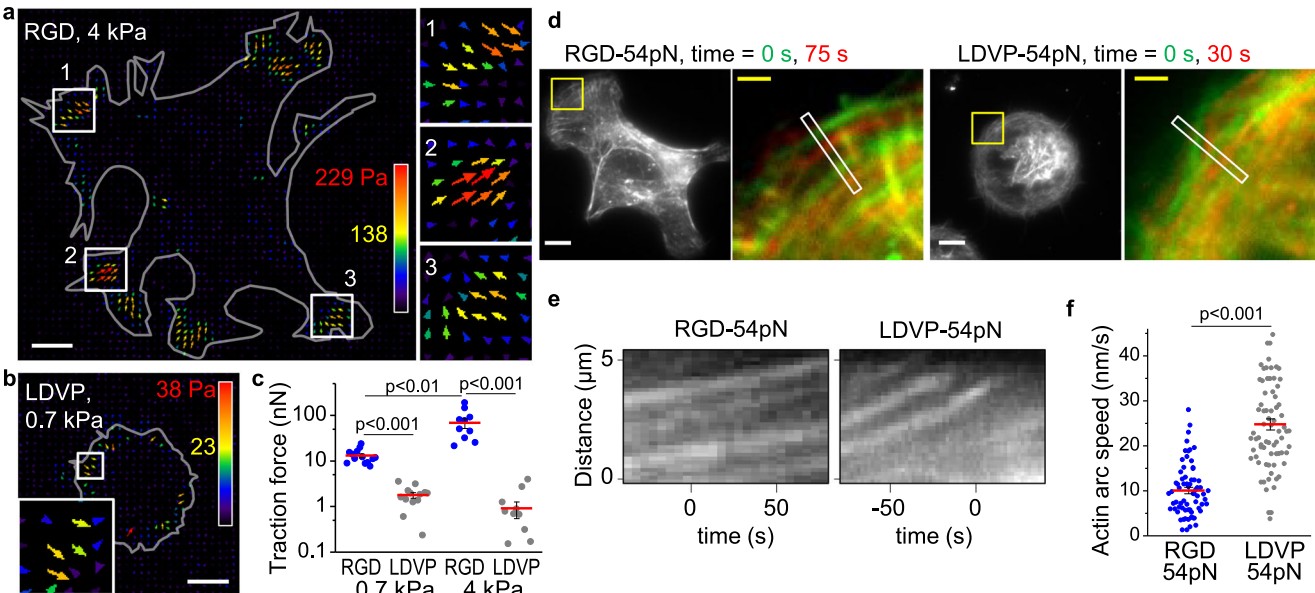

**Fig. 4 | Traction force and retrograde flow rate on RGD-TGT and LDVP-TGT.**
**a–c** Traction force of spreading cells on RGD- or LDVP-coated surface. Cells were seeded on RGD (**a**) or LDVP (**b**) coated polyacrylamide gels for 1 h. Stress vector images are pseudo-colored and length scaled for traction stress magnitude. Gray lines indicate cell outlines obtained from DIC images. Boxed regions are magnified. Scale bars, 10 μm. **c** Total traction force per cell (*n* = 11–13 spread cells). Three independent experiments were conducted. **d–f** Actin retrograde flow rate. **d** Two time points were overlaid for the yellow boxed regions to show actin arc translocation. Scale bars: 10 μm (white), 2 μm (yellow). **e** Representative kymographs obtained from time-lapse images along the white boxed regions shown in **d**. **f** Actin arc speed measured from the kymographs (linear fit; *n* = 64, 71 loci from 15, 13 cells) from two independent experiments. Lines show mean ± SE. Two-sided *t*-test for *p*-values. Source data are provided as a Source Data file.

Retrograde flow of the actin network is counterbalanced by the force exerted by the cytoskeleton on integrin adhesions (Supplementary Fig. 7c, d)[28]. We measured the speed of actin retrograde movement in the leading edge (Fig. 4d, overlayed actin images at two time points in different color) from slopes in kymographs along the direction of actin movement (Fig. 4e, Supplementary Fig. 7e). Retrograde actin flow on RGD-TGT (10.1 nm/s) was more than 2-fold slower than on LDVP-TGT (24.8 nm/s; Fig. 4f), consistent with counterbalancing by the higher molecular force exerted on RGD-TGT and higher bulk traction force measured on RGD-coated surface.

**Molecular tension evolution during cell spreading**
To monitor the dynamics of molecular force exertion during cell adhesion and spreading, time-resolved molecular tension maps were reconstructed by measuring fluorescence signal increase of qTGT between successive image frames[18] (Fig. 5a–h; Movie 5). Early in adhesion, some RGD-54pN rupture was observed near the leading edge with a relatively low frequency. After 25 min, rupture signals became frequent (Fig. 5a). Many instantaneous force signals were punctate (Fig. 5a, yellow inset), but most moved centripetally during the maturation of elongated focal adhesions to form streak patterns (Fig. 5e)[19]. On RGD-33pN, which does not support cell spreading (Fig. 2c), some rupture events were observed over time (Fig. 5d), indicative of abortive spreading attempts; fewer rupture events were seen on RGD-12pN (Fig. 5c; Movie 2).

On the LDVP-33pN surface, significant force signals were detected immediately following cell landing (Fig. 5b, g, i). Punctate signals were observed near the edge of the spreading cell (Fig. 5b). The force signals were often observed under filopodia (Fig. 5b, 14 min; see inset) that appeared outside the cell body, probing the neighboring region. Cumulative tension maps show force signals with protruding patterns near the cell edge at about 30 min when most cells have spread (Fig. 5f, g); force signals continued after spreading was largely completed, suggesting that the adhesions near the cell edge keep testing the surface mechanically. In contrast, rupture of LDVP-54pN was rare (Fig. 5h).

The molecular rate of TGT rupture over time averaged over multiple cells (Fig. 5i) shows that force transmission (>33 pN) through α4β1 occurs starting very early and decreases in frequency in the later stage (after 30 min), in stark contrast to the RGD-binding integrins which showed increased frequency of high force transmission (>54 pN) in later stages.

**Integrin αVβ1 mediates cell spreading on cRGDfK substrates**
Despite its wide use in single molecule force sensors, cRGDfK is a ligand for multiple RGD-binding integrins[29], the contributions of which have not previously been deconvoluted. Identifying a uniquely important RGD-binding integrin would rule out integrin cooperativity in the distinctive behavior on RGD compared to α4β1 substrates. Flow cytometry showed that BJ-5ta cells express RGD-binding integrins α5β1, αVβ3, and αVβ5, and potentially αVβ1, but not α8β1, αIIbβ3, αVβ6, or αVβ8 (Supplementary Fig. 8a). Affinity measurements showed that in the presence of integrin activating Mn²⁺ or as the high affinity, EO conformation in Mg²⁺, integrins α5β1, αVβ1, αVβ3 and αVβ5 bound cRGDfK with 10–100 nM affinities (Fig. 6a, Supplementary Fig. 9).

We collaborated with the Institute for Protein Innovation (IPI) to discover antibodies to integrins α5β1, αVβ3, αVβ5, and αVβ8 from a synthetic yeast-displayed Fab library with a diversity of ~1.4 × 10¹⁰. IPI antibodies and an antibody to αVβ1 from Biogen[30] showed that BJ-5ta cells have high levels of α5β1 and αVβ1 and low levels of αVβ3 and αVβ5 (Fig. 6b).

For inhibition studies we used a different subset of inhibitory IPI and Biogen Fabs with RGD-mimetic heavy chain CDR3 sequences (Supplementary Table 1). Since we used these Fabs to specifically inhibit cell surface expressed integrins, we measured their binding affinities to the target integrins, as well as the other RGD-binding integrins expressed on BJ-5ta cell surface (Fig. 6c and Supplementary Fig. 10a). They show selectivities for αVβ1, αVβ3 and αVβ5 ranging from 160 fold to 2000 fold. Moreover, mAb16 is highly selective to

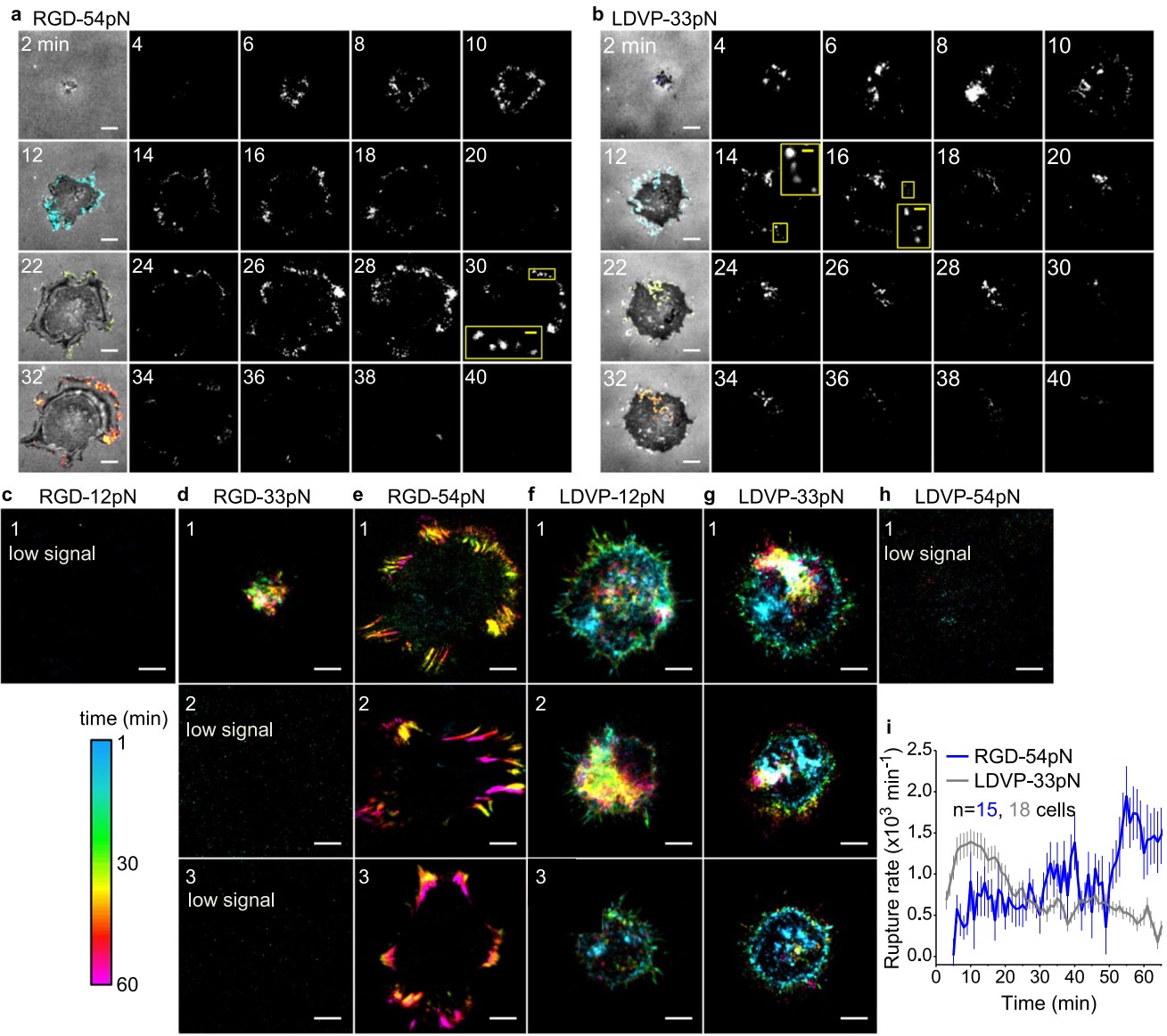

**Fig. 5 | Spatiotemporally resolved TGT rupture.** Time-resolved maps of TGT rupture induced by cell spreading (BHQ2-Cy3). Signal color is keyed by time (see color key). Time-lapse images were taken at 1 min intervals and pixelwise signal change was analyzed. See also Movie 5. The rupture signal, cumulated for a given time period, reflects integrin force transmission events (>T$_{tol}$). Two-minute cumulation is shown for cells spreading on RGD-54pN (**a**) and LDVP-33pN (**b**). Boxed regions are magnified. Panel **a** is at higher contrast than **b** to better show the spatial distribution of signals. Panel **i** directly compare intensity. The rupture signals (colored) were overlaid on RICM images (gray) in the first columns. **c**–**h** Rupture signals superimposed for 1 h for the indicated TGT. Scale bars, 10 μm. Representative results from multiple experiments. **i** TGT rupture rate for multiple cells (mean ± SE). Cells that ruptured less than 200,000 qTGT molecules were analyzed (*n* = 15, 18 cells from 3 independent experiments). Source data are provided as a Source Data file.

integrin α5β1[31] and binds close to the RGD binding site[32]. Each of these four Fabs inhibits binding of RGD ligands to purified ectodomain fragments of the integrins for which they are selective (supplementary Fig. 10b). Fabs at concentrations of 5–10 μM inhibited the desired integrin by 99–100% and other integrins only 0–37% (Fig. 6d). When all three αV integrins were inhibited with a combination of Fabs against αVβ1, αVβ3 and αVβ5, cell spreading was abolished (Fig. 6e); inclusion of α5β1 Fab together with αVβ3 and αVβ5 Fabs gave no significant effect, and α5β1 Fab had no significant effect by itself (Fig. 6f, g). Single or combined inhibition of αVβ3 and αVβ5 did not influence cell spreading on RGD-54pN (Fig. 6f, g). When αVβ3 or αVβ5 inhibition was combined with αVβ1 inhibition, cell spreading was almost completely inhibited. When only αVβ1 was blocked, cells still adhered to RGD-54pN surface, but the spreading was significantly inhibited, as shown by the smaller cell area and lack of elongation (Fig. 6f, g). Thus, αVβ1

was by far the most important integrin for spreading on RGD-TGT substrates. αVβ3 and αVβ5 could each augment the function of αVβ1, but neither was required for full spreading.

The crucial role of αVβ1 in spreading was further illustrated by an alteration of the cytoskeletal morphology upon its inhibition; focal adhesions were not elongated and exhibited a reduced aspect ratio, and alternative types of adhesion structures were formed that were globular in shape, remained near cell edges, and lacked a streaked distribution of paxillin (Fig. 6h, Supplementary Fig. 8b).

We next compared integrins for their ability to transmit high forces and rupture RGD-54pN (BHQ2-Cy3). Rupture events were not suppressed when αVβ3, αVβ5 and α5β1 were blocked; however, TGT rupture was abolished when αVβ1 was blocked (Fig. 6i).

Taken together, these results show that integrin αVβ1 by itself can mediate cell spreading and develop focal adhesions exerting >54 pN

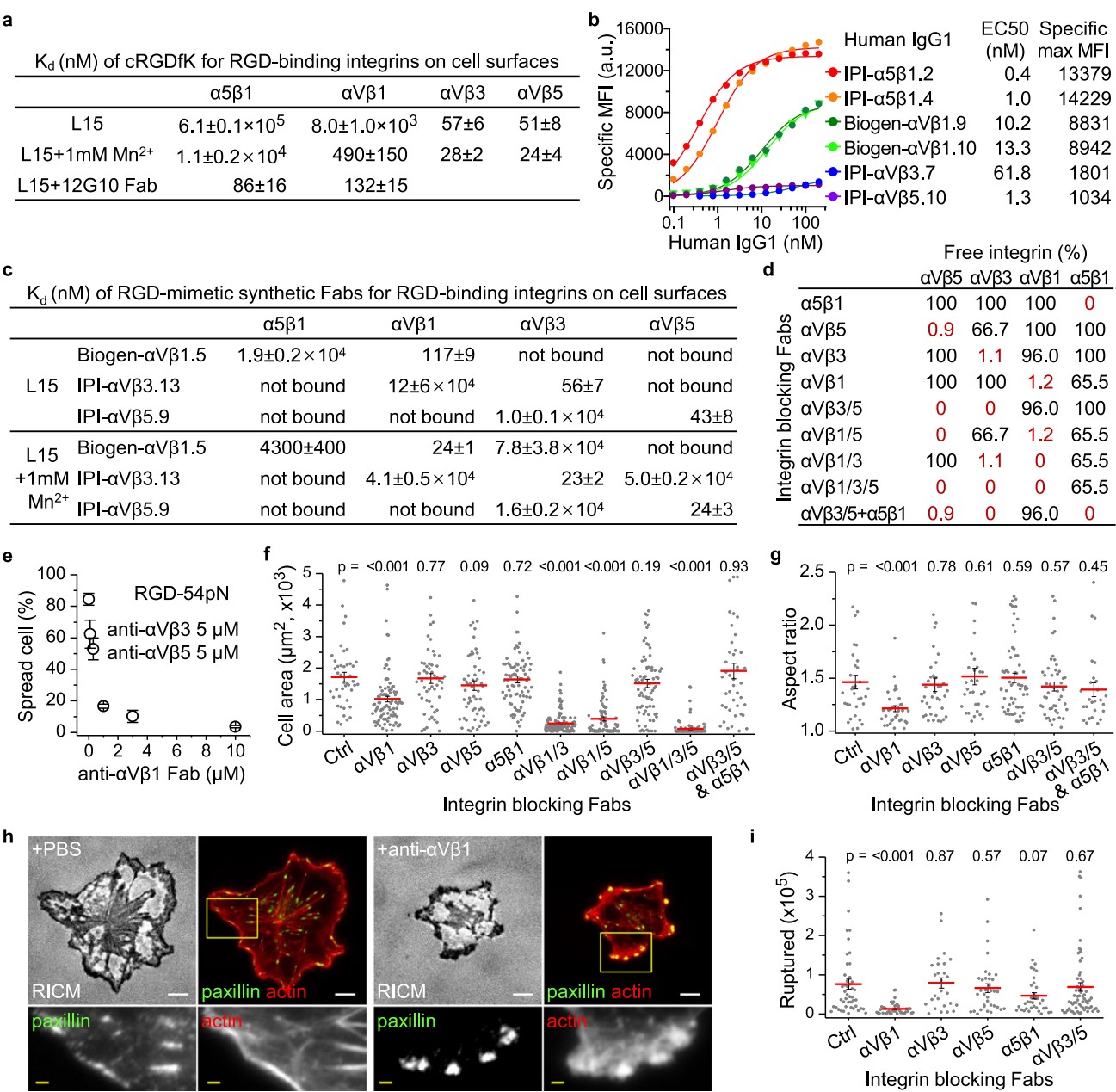

**Fig. 6 | RGD-binding integrin-specific cell spreading and force transmission.**
**a** Binding affinities of cRGDfK peptide to intact RGD-binding integrins on K562 cells in L15 medium with or without 1 mM Mn$^{2+}$ and to the EO states of integrin α5β1 and αVβ1 stabilized by 12G10 Fab. Titration curves are shown in Supplementary Fig. 9. **b** Integrin expression on BJ-5ta cells quantified by dose dependent staining with human IgG1s and fluorescent goat anti-human IgG. Background signal with non-binding IgG1 is subtracted. Binding was fitted to dose response curve to obtain antibody EC50 and maximum specific mean fluorescence intensity (MFI). **c** K$_d$ values of RGD-mimetic Fabs against cell surface RGD-binding integrins. Measurements were either with or without 1 mM Mn$^{2+}$ (Supplementary Fig. 10). Not bound: no significant binding up to 10 μM. **d** Percentage of Fab-unbound RGD-binding integrins in presence of specific inhibiting Fabs or combinations of them. Percentage of Fab-unbound integrin subtype is calculated based on $P^{Fab-unbound} = \sum_i \frac{1}{1 + C_{Fab,i}/K_{d,i}}$, where $C_{Fab,i}$ is the concentration used for $i^{th}$ Fab, and $K_{d,i}$ is the K$_d$ value of ith Fab to the specified integrin subtype. IPI Fabs shown in panel c and mAb16 Fab to α5β1 were at 5 μM; Biogen-αVβ1.5 Fab was at 10 μM.

Affinities of Fabs measured without Mn$^{2+}$ (panel **c**) were used to calculate the percentage of Fab-unbound integrins. **e–i** BJ-5ta spreading in presence of integrin blocking Fabs as described in panel d legend. Cells were pre-incubated with Fabs for 5 min and seeded for 1 h on each surface. **e** Integrin αVβ1 dependent cell spreading on RGD-54pN (mean ± SE for three independent experiments, n = 26–101 cells for each experiment). **f** Close-contact enclosed area of cells on RGD-54pN analyzed from RICM images (mean ± SE; n = 43, 90, 53, 58, 81, 121, 83, 76, 102, 43 cells). **g** Aspect ratio of the area (mean ± SE; n = 33, 34, 37, 32, 61, 48, 26 cells; >1000 μm$^2$). **h** Immunostaining of paxillin (green, AF488) and actin stress fibers (red, SiR-actin). Cells were fixed after 1 h spreading on RGD-54pN. Scale bars: 10 μm (white), 2 μm (yellow). Boxed regions in the top row were magnified in the bottom row. **i** Cells were seeded on RGD-54pN with BHQ2-Cy3 and the ruptured TGT was quantified (mean ± SE; n = 46, 41, 27, 35, 36, 65 cells). Two-sided *t*-test for *p*-values. Images are representative of multiple experiments and cell data points are combined from three independent experiments. Source data are provided as a Source Data file.

tensile force on cRGDfK surface without other RGD-binding integrins. In contrast, integrins αVβ3 and αVβ5, which have higher affinity in their basal ensembles to cRGDfK but lower expression levels, play a supporting role in the early stage of cell spreading. Moreover, integrin α5β1, the best expressed RGD-binding integrin on BJ cells, does not contribute to spreading on cRGDfK, correlating with its low basal ensemble affinity to the cRGDfK peptide (Fig. 6a).

## Spatial distribution and conformational states of integrins

Binding kinetics of Fabs were sufficiently rapid for live-cell imaging of α4 and αV integrins through fluorescent labels on Fabs (Supplementary Fig. 11). On RGD-54pN surfaces, oblong integrin αV clusters were densely located near the cell edge (Fig. 7a). The overlay of αV and actin images (SiR-actin) and the intensity profile along focal adhesion show that αV is connected to the actin bundle in the focal adhesions both near the cell edges and at the end of ventral stress fibers.

In contrast, on LDVP-54pN surfaces, we did not see elongated integrin clusters, thick peripheral actin bundles, or ventral or dorsal stress fibers (Fig. 7b). Instead, discrete small clusters of α4 (Fig. 7b, yellow box), a cross-linked actin network, and dynamic rod-like filopodia were observed near cell edges (Supplementary Fig. 2).

Integrins αV and α4 did not form clusters on LDVP-54pN and RGD-54pN substrates, respectively (Fig. 7c). Paxillin, a cytoskeletal adaptor protein with the most potential binding partners within focal adhesions[33], was present both in the elongated focal adhesions with αV on the RGD surface and in the scattered small adhesions with α4β1 on the LDVP surface (Fig. 7d). Paxillin images compared with ruptured TGTs (LDVP-23pN and RGD-43pN) suggest that force transmission occurred preferentially at the adhesions (Supplementary Fig. 12).

The spatial distribution of low affinity closed (C, BC and EC) and high affinity open (EO) β1 integrin states was also characterized with conformation-specific Fabs. On both RGD-54pN and LDVP-54pN surfaces, active β1 (β1EO) was clustered whereas inactive β1 (β1C) was diffuse (Fig. 7e, f). Therefore, only a fraction of αVβ1 and α4β1 integrins on the ventral cell surface were activated and their distribution suggested they were localized to adhesion structures.

## Tension thresholds for integrin activation are low

To test the hypothesis that the force required to stabilize integrin activation is distinct from that required to stabilize cell spreading, we measured binding of Fab specific for the β1 EO conformation as a function of TGT $T_{tol}$. BJ cells were incubated with biotin-PEG-cholesterol to bring them into close contact with the substrate independently of TGT (Fig. 8a, b). Biotin-functionalization did not interfere with cell spreading on RGD-TGT or LDVP-TGT (Fig. 8b, c).

On RGD-12pN, the close-contact enclosed area, visualized as the dark area in RICM (Fig. 8b), declined relative to the control due to the lower density of biotin binding sites after TGT immobilization (Fig. 8c). While the number of bound 12G10 Fab in this smaller close-contact area was not significantly different from the control (Fig. 8d), 12G10 Fab density significantly increased ($p < 0.001$; Fig. 8e), indicating conformational activation of β1 integrin at the low force on RGD-12pN. On RGD-23pN and RGD-33pN, the contact area increased relative to RGD-12pN, but was not significantly different from the control (Fig. 8c); however, bound 12G10 Fab significantly increased and hence the density of activated β1 significantly increased relative to the control (Fig. 8d, e). On RGD-43pN and RGD-54pN, cell spreading occurred, 12G10 Fab binding also greatly increased (Fig. 8c, d), and 12G10 Fab binding density decreased by 30% compared to that of RGD-33pN probably due to the limited number of integrins for the large cell area (Fig. 8e).

On LDVP-TGT, spreading was significant at all five $T_{tol}$ as was the increase in number of bound 12G10 Fabs (Fig. 8c, d). Furthermore, the density of bound 12G10 Fab and hence activated β1 were significantly above the control and similar at all $T_{tol}$ (Fig. 8c–e). Altogether, these

results show that integrins αVβ1 and α4β1 are activated to the high affinity state by forces that are less than 12 pN and thus that β1 integrin activation does not set the tension threshold of >33 pN for cell spreading on RGD substrates.

β1 integrin activation quantified here by the number of bound AF647-labeled 12G10 Fab (Supplementary Fig. 4a) is interesting to compare to the number of ruptured TGT. As 12G10 Fab (20 nM) diffuses under adherent cells and binds to and dissociates from the EO state of β1 in the time scale of 2–3 min (Supplementary Fig. 11), 12G10 Fab binding after 30 min measures the number of substrate-bound, active β1 integrins at steady state. This number reached ~70,000 integrins per cell for β1 integrins on RGD-54pN and α4β1 on LVDP-54pN (Fig. 8d). As spread area increased with increasing TGT $T_{tol}$, activated integrin density reached plateau levels on each substrate. Importantly, spreading appeared not to be perturbed by 12G10 Fab used here at low concentration as a reporter, because the dependence of spread area on TGT $T_{tol}$ was very similar with and without Fab. In contrast to the steady state value of engaged β1 integrins after 30 min of spreading, TGT rupture continued to accumulate over a 60 min period and reached maximal values of ~200,000 for RGD-43pN and ~100,000 for LDVP-23pN (Fig. 3c). Rupture rate varied with time (Fig. 5i), but assuming that all EO state integrins were resisting cytoskeletal force, and using the number of activated EO state β1 integrins at these forces (Fig. 8d), the average rupture rate over a 60 min period would be ~0.15 and ~0.04 ruptures per activated β1 integrin per minute on RGD-43pN and LDVP-23pN substrates, respectively. Therefore, activated integrins experience large forces in the tens of pN range only infrequently, about once every 7–20 min on average.

## α5β1-mediated cell spreading also requires a high tension

BJ-5ta cells express integrin α5β1 at higher levels than αVβ1 (Fig. 6b). Cell surface α5β1 shows 200-fold higher affinity for cyclic-ACRGDGWCGK (ACRGD)[14,34,35] (Fig. 9a, Supplementary Fig. 13) than cRGDfK (Fig. 6a, Supplementary Fig. 9). We synthesized ACRGD-TGT (BHQ2-Cy3), on which BJ-5ta cells spread efficiently and developed elongated, irregular shapes (Fig. 9b, c). As cyclic-ACRGDGWCGK peptide also binds well to the three αV RGD-binding integrins on BJ-5ta cells (Fig. 9a), we used inhibitory Fabs to test the integrin subtype-dependence of spreading. Blocking α5β1 significantly inhibited cell spreading but blocking αVβ1 or all three αV integrins did not. Spreading was completely abolished by blocking both α5β1 and αVβ1 (Fig. 9b). Thus, adhesion of BJ-5ta cells on ACRGD-TGT is primarily mediated by α5β1 and αVβ1 has a lesser role that is only revealed when α5β1 is blocked. Concordant results were obtained by measuring the aspect ratio of spread cells and ACRGD-54pN rupture (Fig. 9c, d).

The tension requirement for BJ-5ta spreading through α5β1 was tested in the presence of blocking Fabs to αVβ1, αVβ3 and αVβ5. Cells remained spherical and showed little contact on ACRGD-TGT with $T_{tol}$ of 12, 23, 33 pN and spread well on ACRGD-TGT with $T_{tol}$ of 43 and 54 pN (Fig. 9e, f). Integrin α5β1-mediated ruptures of ACRGD-43pN and ACRGD-54pN showed streaks, at the outermost regions of spread cells, suggestive of the high force transmission events in focal adhesion maturation (Fig. 9e, g). These results showed that α5β1 resembled αVβ1 in its ability to transmit high single molecule tension forces, mediate asymmetric cell spreading and focal adhesion formation, and requirement for force levels >33 pN to support cell spreading.

## Discussion

We discovered here that in the same cell, the RGD-binding integrins αVβ1 and α5β1 differ dramatically from α4β1 integrin in mechanotransduction, i.e. in the types of cytoskeletons they assemble, the forces those cytoskeletons transmit, and in the way in which these integrins respond to and regulate force transmission. Integrin α4β1 is expressed on many types of mesenchymal cells, such as fibroblasts, and also on certain white blood cells such as lymphocytes but not

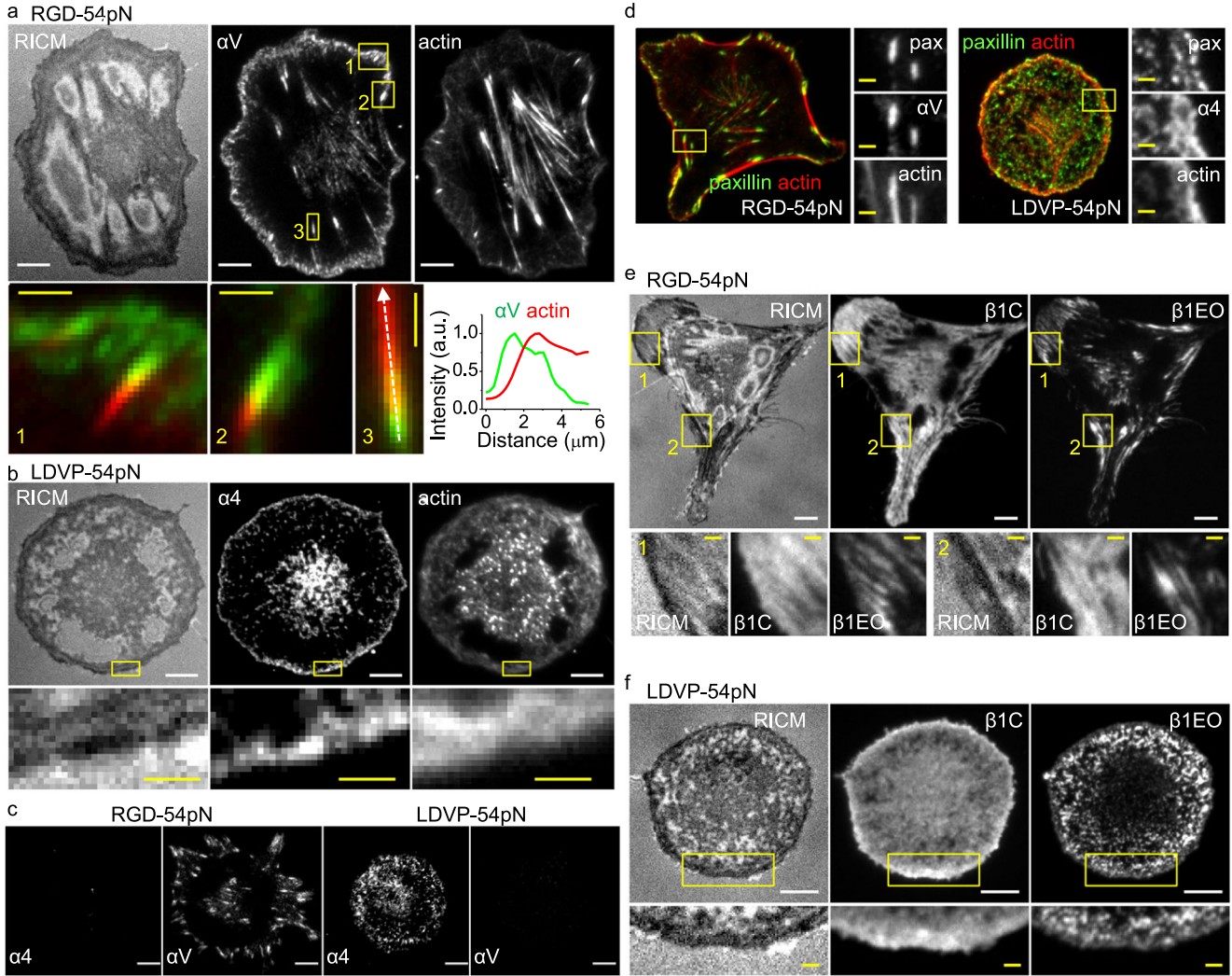

**Fig. 7 | Integrin-based subcellular adhesion structures on RGD and LDVP surfaces.** Fab (100 nM for HP1/7 and 20 nM for the others) and Sir-actin (10 nM) were added 10 min before live-cell imaging of cells seeded for 1 h. **a** Integrin αV (AF488-13C2 Fab, green in the merged images) and actin stress fibers (SiR-actin, red in the merged images) on RGD-54pN. Intensity profiles (bottom right panel) are along the dashed line in the third enlarged image. Arbitrary units (a.u.). **b** Integrin α4 (AF488-HP1/7 Fab) and actin stress fibers (SiR-actin) on LDVP-54pN. **c** Integrin α4

(Cy3-HP1/7 Fab) and αV (AF647-13C2 Fab). **d** Paxillin (AF488, green) immunostaining of fixed cells imaged with SiR-actin (red) and αV (Cy3-13C2 Fab) or α4 (Cy3-HP1/7 Fab). **e** Extended open β1 (Cy3-12G10 Fab) and closed β1 (AF647-mAb13 Fab) on RGD-54pN. **f** Extended open β1 (Cy3-12G10 Fab) and closed β1 (AF647-mAb13 Fab) on LDVP-54pN. Boxed regions are magnified in the bottom row or right column. Scale bars: 10 μm (white), 2 μm (yellow). Source data are provided as a Source Data file.

neutrophils. Early studies of α4 subunit chimeras examined their function in K562 and CHO cell transfectants. Compared to chimeras containing α4 cytoplasmic domains, chimeras lacking cytoplasmic domains and those with α5 and α2 cytoplasmic domains favored firm adhesion over rolling adhesion, greater spreading, greater association with focal adhesions, and lesser cell migration[36]. Another study compared α4 and α5-transfected cells and showed formation of cortical actin and more migration with α4 compared to actin in stress fibers with α5[37]. Chimeras with αIIb similarly showed that the α4 cytoplasmic domain antagonized spreading and that distinct biological responses were due to binding of paxillin to α4[38], which required binding to cytoplasmic domain residues Glu-983 and Tyr-991 and was blocked by phosphorylation at intervening residue Ser-988[39,40]. In this study, we show distinctive properties of RGD-binding versus α4 integrins observed through distinct actin and paxillin cytoskeleton architectures, rounder cells with α4 integrins, lack of cell migration mediated by α4β1 integrins in BJ-5ta cells, and the different threshold tension required for spreading and the magnitude of force exertion through αVβ1 and α5β1 versus α4β1 integrins.

Previously, contributions of individual RGD-binding integrins in single-molecular force sensor studies have not been isolated using blocking reagents. Utilizing RGD-mimetic antibodies against specific αV integrin heterodimers, obtained from synthetic yeast-displayed Fab libraries, together with mAb16 to α5β1, we selectively inhibited each of the four RGD-binding integrins expressed on BJ-5ta cells with Fabs, which excluded artifacts from the avidity effect of IgGs. We found that integrin αVβ1 has the major role in BJ-5ta fibroblast spreading on cRGDfK substrates. Only blocking αVβ1 significantly inhibited spreading, development of single molecule forces >54 pN (ruptures were reduced by ~6-fold), development of cell asymmetry, and focal adhesion formation. Blocking αVβ1 also decreased focal adhesion length and changed the distribution of paxillin and actin. αVβ1 is a fibronectin-binding integrin and is activated during development by binding to fibronectin in somitogenesis, along with α5β1 but not αVβ3 or αVβ5 integrins[41–43].

The Fabs characterized here for integrin selectivity should be useful in the future for working out the individual roles of RGD-binding integrins in the many different types of mechanosensing systems in

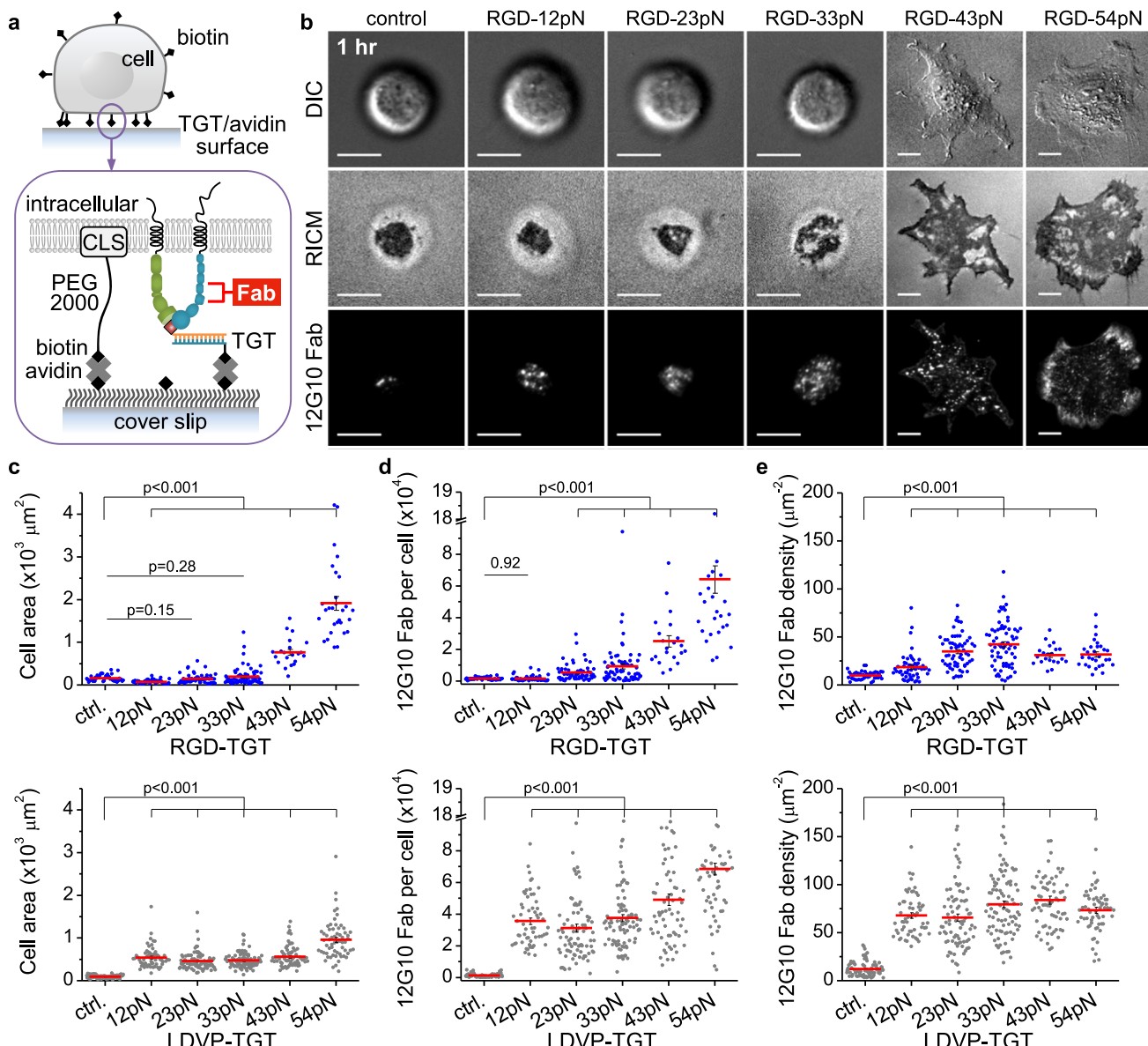

**Fig. 8 | Ligand-bound integrins are activated to the EO state at lower force than required for spreading on RGD. a** Schematic of immobilization of biotin-functionalized cells for activated integrin β1 detection. Cholesterol (CLS) linked to biotin with PEG (~13 nm) immobilizes cells on neutravidin and TGT-coated surfaces. **b** Cell images and AF647-12G10 Fab (20 nM) signal 1 h after seeding. Scale bar, 10 μm. **c**–**e** Cell area and 12G10 Fab signal were measured 30 min after seeding (*n* = 43, 55, 62, 69, 21, 29; 83, 59, 83, 92, 68, 62 cells from two independent experiments). **c** Close-contact enclosed area of cells. Cells did not spread on the control or 12–33 pN RGD-TGT surfaces. **d** The number of 12G10 Fab counted per cell. Fluorescent Fab signal was calibrated by analyzing stepwise signal increase due to nonspecific Fab binding to surfaces outside the area imaged with cells (Supplementary Fig. 4a). **e** Density of 12G10 Fab for each cell. Lines show mean ± SE. Two-sided *t*-test for *p*-values. Not significant (ns), *p* > 0.05. Source data are provided as a Source Data file.

which these integrins have been studied. On RGD single molecule force-sensor substrates, αVβ3 is localized to regions of high tension, whereas α5β1 localizes to both high and low force regions[44]. Comparisons on fibronectin substrates of fibroblasts genetically deficient in either the integrin αV or β1 subunits showed that αV integrins accumulate in high traction force areas, mediate rigidity sensing and develop large focal adhesions, whereas β1 integrins generate larger traction forces[45]. However, that study could not study αVβ1 integrin's contribution to high traction forces because αVβ was not present among the αV integrins in β1-deficient cells or the β1 integrins in αV-deficient cells. We found that αVβ3 and αVβ5 when combined were capable of mediating spreading but by themselves were much less effective than αVβ1 on cRGDfK peptide conjugated TGT. α5β1 was also capable of mediating asymmetric cell spreading and high tension force

transmission by itself on TGTs conjugated to cyclic-ACRGDGWCGK peptide. The similarity in mechanotransduction in BJ-5ta cells between the RGD-binding integrins, αVβ1 and α5β1, markedly contrasted with that of α4β1.

We found that a single type of cell, i.e. BJ-5ta dermal fibroblasts, can adopt markedly different shapes and assemble different types of actin cytoskeletons in response to engagement of ligands outside the cell through RGD-binding integrins and through α4β1. The response of the cytoskeletal machinery is thus not predetermined by cell type, but is highly regulatable by the type of ligand outside the cell. The ligand, in turn, then determines the subtype of integrin that engages to the cytoskeleton and initiates distinct downstream signaling that regulates the complement of actin regulatory proteins activated to mediate the assembly of distinct actin architectures and dynamics. Moreover,

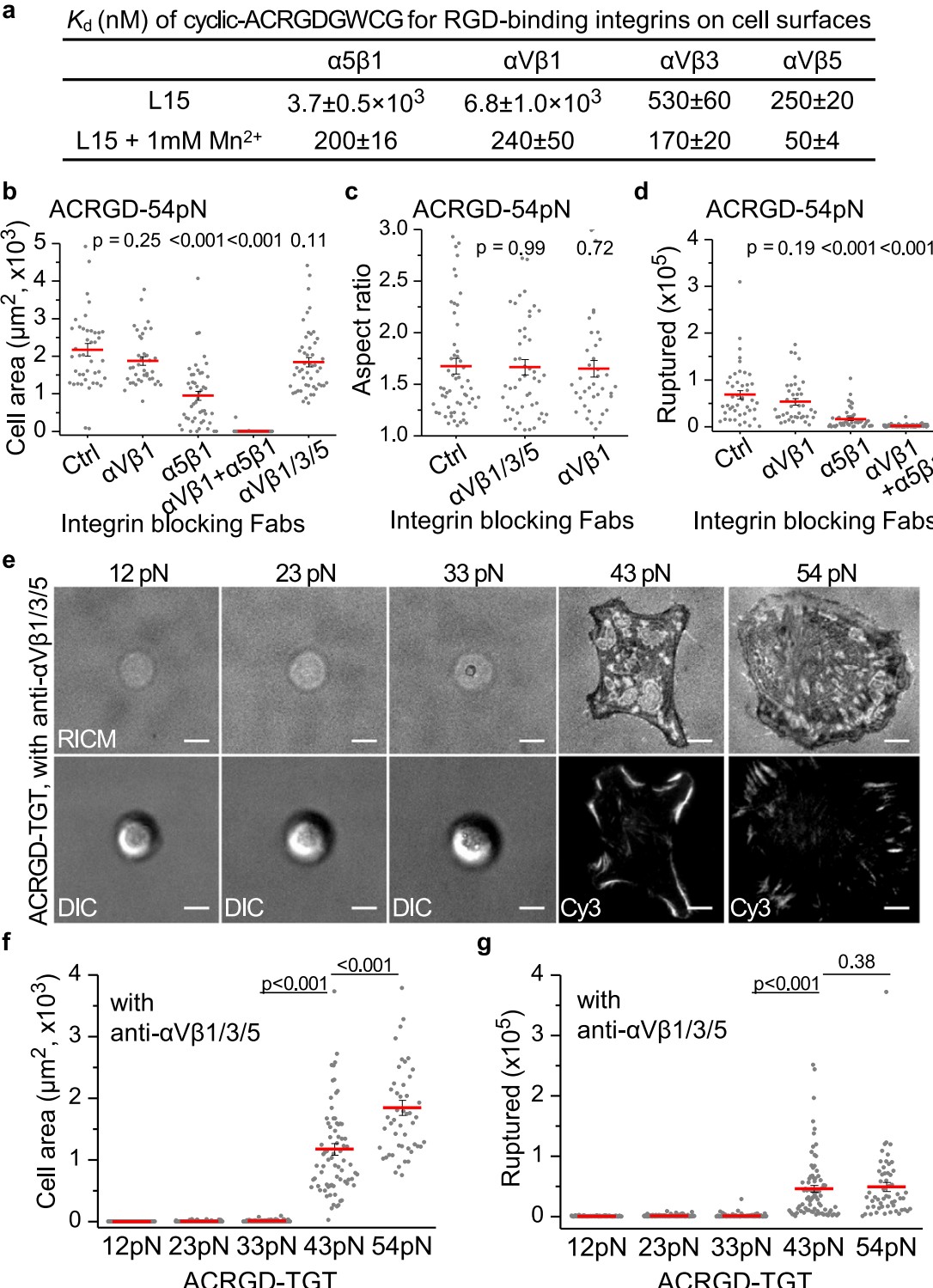

**a** $K_d$ (nM) of cyclic-ACRGDGWCG for RGD-binding integrins on cell surfaces

|  | α5β1 | αVβ1 | αVβ3 | αVβ5 |
|---|---|---|---|---|
| L15 | $3.7\pm0.5\times10^3$ | $6.8\pm1.0\times10^3$ | $530\pm60$ | $250\pm20$ |
| L15 + 1mM Mn$^{2+}$ | $200\pm16$ | $240\pm50$ | $170\pm20$ | $50\pm4$ |

**Fig. 9 | Integrin α5β1 mediated cell spreading. a** Affinity of ACRGD for integrins on intact cells. Titration curves are shown in Supplementary Fig. 13. **b**–**d** Behavior of BJ-5ta on ACRGD-54pN (BHQ2-Cy3) surfaces after 1 h in presence of integrin blocking Fabs described in Fig. 6 legend. **b** Close-contact enclosed area from RICM images ($n = 43, 38, 47, 96, 50$ cells). **c** Aspect ratio of contact area ($n = 56, 49, 38$ cells, $>500\,\mu m^2$). **d** Ruptured TGT counts ($n = 44, 40, 48, 96$ cells). **e**–**g** Cell spreading on ACRGD-TGT of different $T_{tol}$ in presence of integrin blocking Fabs. **e** Representative images of cells with RICM (top row) and same cells with DIC

or Cy3 fluorescence to show TGT rupture (bottom row). Dim RICM rings (-20 μm diameter) with 12, 23 and 33 pN TGT are the shadows of the spherical cell body, not the close contacts. Scale bars, 10 μm. **f** Close-contact enclosed areas ($n = 94, 95, 79, 75, 50$ cells). **g** Ruptured TGT counts ($n = 94, 96, 88, 81, 55$ cells). Lines show mean ± SE. Two-sided $t$-test for $p$-values. Images are representative of multiple experiments and cell data points are combined from two or more independent experiments. Source data are provided as a Source Data file.

these different cytoskeletal architectures correlated with differences in the tension threshold for spreading and the forces transmitted through the integrins to their ligands. Integrin α4β1 mediated spreading at the lowest $T_{tol}$ tested here, 12 pN, whereas multiple RGD-binding integrins together, and the individual αVβ1 and α5β1 integrins, required $T_{tol}$ between 33 and 43 pN for spreading. The affinities of the ligands used in this study for α4β1, αVβ1 and α5β1 in the extended-open conformation, which represents the catch-bond force-activated state[46], were 0.15 ± 0.05 nM (LDVP to α4β1)[22], 132 ± 15 nM (cRGDfK to αVβ1) and 2.2 ± 0.5 nM (cyclic-ACRGDGWCGK to α5β1)[14], respectively. No relationship is evident between the different $T_{tol}$ required for α4β1, αVβ1, and α5β1 for spreading on their cognate ligands and affinity for ligand.

Paxillin is a multi-domain scaffold recruiting numerous regulatory and structural proteins that together control dynamic changes in cell adhesion and cytoskeletal reorganization[47]. Quantitative mass spectrometry for integrin adhesion complexes suggested that LIM domain proteins, including paxillin, are potential tension sensors[48]. In fibroblasts, paxillin shows high degree of spatial correlation with RGD-binding integrin force[44] and directly binds to kindlin, which is crucial for activating β1 integrins and adhering to fibronectin[49]. Even though we observed that paxillin was recruited in both α4β1 and RGD-binding integrin mediated adhesions, the difference in mechanosensing and force transmission implies that the paxillin signaling may work in different manners. Unlike αIIb, α5 and β1 tails, the α4 tail binds tightly to paxillin, which regulates cell spreading[38,40]. This direct paxillin interaction may underlie the low tension threshold for α4β1-mediated cell spreading and its distinctive cytoskeleton architecture.

Importantly, our results ruled out the hypothesis that the tension required for integrin activation determines the tension required for integrin-dependent spreading on substrates. Using fluorescently labeled Fab specific to activated β1 in live cells, we found that integrin β1 molecules were activated on both LDVP-12pN and RGD-12pN surfaces. On LDVP substrates where spreading occurred at all $T_{tol}$ values, integrin β1 activation densities were comparable at all $T_{tol}$ values. Although activation was highly significant on RGD-12pN surfaces, the density of activated integrin β1 molecules was greater on RGD-23pN and still greater on RGD-33pN surfaces. The increasing integrin β1 activation on 12, 23, and 33 pN RGD substrates was consistent with a similar increase in the number of TGT ruptures. Overall, our findings that β1 integrin molecules are stabilized in the EO state at forces of less than 12 pN confirmed a suspicion based on measurements of average forces on integrin LFA-1 of ~1.5 pN[15] and predictions of the tensile force required for integrin activation based on measurements of the free energy difference between the BC and EO states of ~2.5–4 kcal/mol for integrins α4β1 and α5β1 on cell surfaces[22]. Considering the increase in distance between the ligand binding site and the C-terminus of β-tail domain for integrin conformational change from the BC to the EO state (~14.5 nm), a force of 1.2–1.9 pN gives an energy that stabilizes 50% of the integrins in the EO state; forces of 1.9–2.6 pN and 2.6–3.3 pN stabilize 90 and 99% of the integrins in the EO state, respectively.

Forces of a few pN across RGD-binding integrins have indeed been detected using other force sensors including peptide springs[9,44,50,51]. The peptide spring sensors using FRET measure real-time integrin forces but only in the low force regime (<7 pN)[9]. Forces higher than the detection maximum are not quantifiable as a result. In contrast, our TGTs detect or modulate integrins transmitting higher force (12–54 pN). High force transmission events would be relatively rare and transient, so most integrins would be under low tension at a given time even if they are engaged with the ligand. Because TGTs can record relatively infrequent high force transmission events, we can quantify the cumulated number of the events (1 h cumulation in Fig. 3; 1–2 min cumulation in Fig. 5). Indeed, based on the steady state number of activated integrins bound to RGD-43pN, we estimated in Results that integrins experience high forces only once every seven to twenty

minutes, and therefore experience low forces at most timepoints. Thus, the results with both types of sensors suggest that at any given time, most integrins experience low force. The unique contribution of TGT studies is to show that the relatively infrequent high integrin force events play an important role in regulating cell spreading. Furthermore, we have found that when RGD-12pN and RGD-54pN TGT were mixed on substrates, substantial RGD-12pN rupture occurred in transient (<30 s in duration) puncta where no RGD-54pN rupture was observed[18]. Therefore, integrins may be activated during initial probing of substrate rigidity.

What then explains the requirement for force between 33 and 43 pN for spreading on RGD substrates? The differences in force required for spreading mediated by α4β1 integrin and RGD-binding integrins correlated with different levels of force that the cytoskeleton exerted on these integrins. TGT rupture was most frequent on LDVP-23pN for α4β1, on RGD-43pN for αVβ1, and similar on ACRGD-43pN and ACRGD-54pN for α5β1. Actin generates force through polymerization or through coupled myosin motors. Cytoskeletons assembled by both subtypes of integrins required actin polymerization for cell spreading and TGT rupture. As shown by inhibition with para-amino-Blebbistatin, myosin II had no role in LDVP-12pN rupture but was required for about one third of LDVP-33pN rupture and half of RGD-54pN rupture. These results suggest that the RGD-engaged cytoskeleton either matures to exert forces of >33 pN, and thus breaks TGT with $T_{tol}$ of ≤33 pN before lamellipodium spreading can be stabilized, and/or disassembles if forces in this range cannot be stabilized. The results with LDVP-12pN and LDVP-33pN rupture, on the other hand, suggest that the cytoskeleton assembled by α4β1 on LDVP-12pN senses low rigidity and thus limits force application by actomyosin. In relation to these findings, non-muscle myosin II is dispensable for the assembly and disassembly of nascent adhesions inside the lamellipodium[52,53] and actomyosin contraction generates high forces (>54 pN) through single integrin-ligand bonds in mature focal adhesions[19]. We speculate that the high force resistance required for spreading on RGD and the higher forces transmitted through RGD-binding integrins may be a specialization that allows these integrins to interrogate the cellular environment to find the stiffest locations for cell anchoring in the process known as durotaxis[54].

Overall, the results suggest that the $T_{tol}$ requirement for spreading is set by the cytoskeleton, and that the different force requirements for spreading are a consequence of the different levels of force exerted by the RGD and LDVP-engaged cytoskeletons and differences among their cytoskeletal machineries in ability to sense substrate stiffness and adjust force exertion accordingly. Important studies in the field of mechanotransduction have shown that substrate stiffness regulates cell differentiation[55,56]. Our study shows that integrin α4β1 and RGD binding integrins including αVβ1 and α5β1 transmit quite different types of signals into cells. In this case, it appears to be the chemical differences between integrin subtypes and presumably their membrane and cytoplasmic domains that make the differences. Both integrin subtype-specific signaling and substrate stiffness-dependent signaling must work together in vivo.

## Methods

### Selecting integrin selective antibodies from a synthetic yeast-displayed Fab library

cDNAs of human RGD-binding integrin ectodomains (including αVβ1, αVβ3, αVβ5, αVβ6, αVβ8, α5β1, α8β1, and αIIbβ3) with N-terminal secretion peptide, followed by HRV3C digestion site on both subunits, C-terminal clasp[57] and C-terminal detection tags (HA tag on β subunit, protein C tag on α subunit), and purification tags (His tag on β subunit, and twin strep tag on α subunit) were produced in transiently transfected Expi293F cells and purified from culture supernatant by Ni-NTA affinity purification followed by size-exclusion chromatography either

directly following affinity purification (clasped ectodomain) or after HRV3C digestion (unclasped ectodomain).

We screened for integrin-specific antibodies using the Fab library developed at Institute for Protein Innovation (IPI) containing $1.4 \times 10^{10}$ unique Fab sequences, each displayed on the surface of Saccharomyces cerevisiae yeast cells. The library was enriched for yeast clones displaying integrin-specific Fab with two rounds of magnetic-activated cell sorting (MACS) employing streptavidin-coupled magnetic beads coated with biotinylated integrin unclasped ectodomains. Using fluorescence-activated cell sorting (FACS), yeast was next subjected to five alternating rounds of positive selection with target integrin ectodomains (FACS1 and FACS3), with poly-specificity reagent (PSR); i.e., biotinylated detergent lysate of baculovirus-infected Sf9 membrane proteins (FACS2), and negative selection against untargeted integrins (FACS4 and FACS5). For example, with αVβ3 integrin, in FACS1 and FACS3, cells were labeled with 100 nM biotinylated unclasped integrin αVβ3 ectodomain. In FACS2, cells were labeled with 100 nM biotinylated PSR reagents. In FACS4 and FACS5, cells were labeled with 100 nM biotinylated unclasped integrin αVβ3 ectodomain and 100 nM each of αVβ1, αVβ5, αVβ6, αVβ8, α5β1, α8β1 and αIIBβ3 ectodomain in unbiotinylated clasped form using PE-labeled streptavidin and Alexa Fluor 647 labeled 12CA5 antibody to the C terminal HA tag, and selected positively for biotin and negatively for the HA tag.

The top-ranked sequences from next-generation sequencing after FACS5 were down-selected to 13 for DNA synthesis and recombinant expression as IgG1 in Expi293F cells. Protein A-purified antibodies were used at 50 nM for immunofluorescent staining on K562 stable transfectants of each RGD-binding integrin in HBSS buffer containing 0.2 mM $Ca^{2+}$, 1 mM $Mn^{2+}$, and 1% BSA. Antibodies showing good binding to target integrin transfectants and minimal binding to other integrin transfectants were then characterized by dose-dependent immunofluorescent staining on target K562 stable transfectants in HBSS buffer containing 1 mM $Ca^{2+}$, 1 mM $Mg^{2+}$ and 1% BSA and characterized.

## Antibodies and preparation of Fabs

Hybridomas were 10E5[58], 12G10[59], 17E6[60], 7.1G10[61], HP1/7[62], LM609[63], mAb13[31], mAb16[31], and P1F6[64]. IgG produced from hybridoma was purified by protein G. The following antibodies were recombinantly expressed in Expi293F cells (Gibco, A14527) cultured in Expi293 medium (Gibco, A1435102) and transfected using FectoPRO (Polyplus, 101000014). Valproic acid and glucose were added to the culture 24 h after transfection to final concentrations of 3 mM and 0.4%, respectively. 13C2[65] antibody amino acid sequence was determined by REmAb (Rapid Novor). ADWA2[66] was recombinantly expressed as IgG2a. Three antibodies to αVβ1[30], Biogen-αVβ1.5 (SEQ ID NO:35 and 22), Biogen-αVβ1.9 (SEQ ID NO:61 and 58) and Biogen-αVβ1.10 (SEQ ID NO:64 and 58) were Exemplary Antibodies 5, 9, and 10, respectively, in the patent, and were expressed as IgG1. MAB6194 was from R&D Systems. All Fabs were generated with papain (100:1 antibody-papain mass ratio) in PBS with 10 mM EDTA, 10 mM L-Cysteine, at 1 mg/mL antibody concentration for 1 h at 37 °C. 30 mM iodoacetamide was added to deactivate papain. After buffer exchange, Fabs were purified with Hi-Trap Q chromatography in 50 mM Tris-HCl pH 9.0 with a gradient in the same buffer to 0.5 M NaCl. Fluorescently labeled Fabs were made by conjugating with either Alexa Fluor 488 or Alexa Fluor 647 NHS Ester (Thermo Fisher Scientific, A20000 or A20006) to lysine side chains in PBS and molar ratio of dye to Fab was controlled to be in the range of 1–2.

## Integrin surface expression level by immunostaining

Surface expression of integrin αV-, α4-, α5-, α8-, β1-, β5-, β6-, and β8-subunits, and integrin αVβ3 and αIIBβ3 heterodimers on BJ-5ta cells was checked by immunostaining with mouse IgGs with various isotypes, and the expression levels of α5β1, αVβ1, αVβ3 and αVβ5 on BJ-5ta cells were quantified by immunostaining with human IgG1 antibodies.

Cells ($10^6$/mL in L15 supplemented with 1 mg/mL BSA) were incubated with 7.5 µg/mL primary mouse IgGs or indicated concentrations of human IgGs on ice for an hour, followed by 3 washes with cold PBS, and then incubated with 2 µg/mL AF647-conjugated goat anti-mouse IgG or 5 µg/mL APC-conjugated goat anti-human IgG for 30 min on ice, followed by 3 washes with cold PBS, and subjected to flow cytometry (BD, FACSCanto II). FlowJo (Tree Star, version 10.7.1) was used for flow cytometry data analysis. For dose-dependent staining with human IgG1, the background-subtracted specific Mean Fluorescent Intensity (MFI) at indicated concentration of primary IgGs was fitted with dose-response for EC50 and the maximum specific MFI using Prism (GraphPad Software, version 9).

## Tension tolerance of TGT

Tension gauge tethers (TGTs) with 18 bp DNA duplex region were used[19]. The sequence of duplex region is GGC CCG CAG CGA CCA CCC (ligand conjugated strand). The nominal values of TGT tension tolerance were used as estimated by previous reports[11,19]. In brief, DNA rupture force was estimated based on the model formulated by de Gennes: $T_{tol} = 2f_c[\kappa^{-1} \tanh(\kappa l/2)]$ where $f_c$ is breaking force per base pair (3.9 pN), $l$ is the number of base pairs, and $\kappa^{-1}$ is an adjustment length (6.8 bp)[67]. The parameters were obtained from DNA rupture experiments using magnetic tweezers (stepwise force increment, $\Delta F = 2$–10 pN and $\Delta t = 1$ s)[68]. The tension tolerance of each TGT was not experimentally measured and the value depends on loading rate which is a measure of how quickly applied force increases. Because the physiological loading rate for integrin-ligand interactions is unknown and would vary, the tension tolerance values should be considered an approximation and the relative comparison between TGTs is important. The estimation should be close to the true values when the loading rate is lower than ~10 pN/s because single-molecule optical and magnetic tweezers studies have shown that DNA unzips at 10–15 pN forces (compared to TGT-12pN, unzipping configuration) and long DNA molecules start to melt at ~60 pN (compared to TGT-54pN, shearing configuration). Recent magnetic tweezers experiment also validated the estimated tension tolerance values of TGT variants (45 and 56 pN) at loading rate of 1 pN/s[27].

## TGT conjugated with peptidomimetic ligands

TGT was assembled by annealing two complementary DNA oligos (18 bp), a ligand strand and an immobilization strand with five different biotin linker positions. To make the ligand strand of RGD-TGT, a cilengitide analog, cRGDfK, with a spacer attached to its lysine side chain, cyclo [Arg-Gly-Asp-D-Phe-Lys(8-amino-3,6-dioxaoctanoic acid dimer)] (Vivitide, PCI-3696-PI, Supplementary Fig. 14), was used. The terminal free amine on the spacer was reacted with the N-hydroxysuccinimide (NHS) ester group of sulfo-SMCC (Thermo Fisher Scientific, 22622). The maleimide group of sulfo-SMCC was subsequently conjugated to 3' thiol modified DNA oligonucleotides as described[19]. For the ligand strand of LDVP-TGT, the 2-methylphenylureaphenylacetyl (MUPA) moiety with LDVPAAK peptide (MUPA-LDVPAAK; now commercially available, Tocris, 7020, Supplementary Fig. 14) was custom synthesized by Bio-Synthesis and the amine group of its lysine side chain was reacted with the NHS group of sulfo-SMCC, and subsequently conjugated to DNA oligos with 3' thiol modifications, as described for the ligand strand of RGD-TGT. To make the ligand strand of ACRGD-TGT, a cyclic-ACRGDGWCGK peptide with a spacer, 8-amino-3,6-dioxaoctanoic acid dimer attached to its lysine side chain (equivalent to cRGDfK product) and a disulfide bond between its cysteines was custom synthesized by Vivitide (Supplementary Fig. 14). To preserve the integrity of the disulfide bond in the peptide, we chose not to use the sulfo-SMCC crosslinking strategy, as our protocol used reducing reagents. We therefore used DBCO (dibenzocyclooctyne) and azide ligation strategy[69]. Briefly, DBCO-NHS (Lumiprobe, 34720) was linked to the terminal free amine on the

spacer of cyclic-ACRGDGWCGK, in DMSO (5% triethylamine). In parallel, Azide-NHS (Lumiprobe, 53720) was linked to 3' amine-modifed DNA oligo at pH 8.5. Azide-modified DNA oligo and DBCO-modified cyclic-ACRGDGWCGK peptide were reacted at pH 7.3 (HEPES 20 mM) to form the peptide conjugated DNA oligo. The peptide and DNA oligo was purified by reverse phase HPLC purification (solvent A: 0.05 M triethylammonium acetate, solvent B: acetonitrile) after each step. The ligand strand DNA oligos were also labeled with a fluorescent probe (Cy3 or Atto647N) or with a quencher (BHQ2) at their 5' ends.

For the immobilization strand, an amine-modified DNA oligo was reacted with NHS-PEG12-biotin (Thermo Scientific, 21312) or a (dT)6 linker was inserted between the biotin and the duplex portion. The linkers were added to minimize steric hindrance by the surface after immobilization on neutravidin. A fluorescent probe (Cy3 or Atto647N) was labeled at 3' for qTGTs. DNA oligos were HPLC purified after each chemical reaction. BHQ2-conjugated DNA oligos were purchased from Biosearch Technologies (double HPLC purification) and the others were from Integrated DNA Technologies (HPLC purification). See Supplementary Table 2 for a full description of DNA sequences and modifications.

## Surface passivation and functionalization

Glass coverslips were densely PEGylated to minimize non-specific interactions such as the binding of cell-secreted extracellular matrix proteins. Coverslips cleaned by piranha etching solution (a 3:1 mixture of sulfuric acid and hydrogen peroxide) were incubated in silanization solution (UCT, 1760-24-3; 2% aminosilane and 5% acetic acid in methanol) for 30 min and reacted with a mixture of Polyethylene glycol (PEG) with and without biotin (1:19, 150 mg/ml; Laysan Bio, Biotin-PEG-SVA-5000 and MPEG-SVA-5000) for 4 h. Surfaces were washed thoroughly and dried using nitrogen gas after each step. A 3D-printed dish (PLA) or a silicon gasket (Grace Bio Labs, 103280) was attached to the coverslip to form a dish. Neutravidin (Thermo Fisher scientific, 31000; 0.4 mg/ml) was added to the dish for 10 min and washed out thoroughly. The dish was emptied and a droplet (3 μL) of TGT solution (100 nM unless specified otherwise) was placed on the coverslip for 10 min in a humid condition to locally immobilize TGT. For the lateral drift correction for time-resolved TGT rupture analysis, biotin-coated gold nanoparticles (Luna Nanotech, GNP-BIOT-100-2-04; 100 nm) were sparsely immobilized as fiducial markers. For protein ligand coating, protein (50 ug/ml in PBS) was incubated in glass bottomed imaging dishes for 1 h at RT without additional passivation.

## Cell culture

Cells were maintained in medium supplemented with 1X antibiotic-antimycotic (Gibco, 15240062), and 10% fetal bovine serum (Corning, 35-011-CV) in 5% CO2 at 37 °C. Human foreskin fibroblasts immortalized with hTERT (BJ-5ta, ATCC® CRL-4001™) were cultured in Dulbecco's Modified Eagle's Medium (Sigma-Aldrich, D5796). Human Lymphoblast K-562 (ATCC® CCL-243™) were cultured in RPMI1640 medium (Gibco, 11875093) additionally supplemented with 3 mg/L puromycin (Sigma, P8833). Expi293F cells (Gibco, A14527) were cultured in Expi293 medium (Gibco, A1435102).

## Live cell imaging

The cells were gently harvested by washing in PBS for 1 min, incubating with 0.05% trypsin for 3 min, and then neutralizing with cell culture medium (FBS 10%). The cells were spun down and resuspended with Leibovitz's L-15 Medium (Gibco, 21083027) supplemented with 2 g/L glucose, then seeded on the TGT surface at an approximate density of 30 cells/mm$^2$. For integrin inhibition experiments, cells were incubated with Fab for 5 min before seeding. For cell immobilization experiments, cells were incubated with 1 nM Cholesterol-PEG-Biotin (Nanosoft Polymers, 3084-2000) for 5 min, followed by washes by centrifugation to remove biotin in solution before seeding. The

imaging area was humidified and the temperature was maintained at 37 °C. A Nikon Eclipse Ti microscope equipped with Xenon arc lamp, Nikon perfect focus system, custom total internal reflection fluorescence microscopy (TIRFM) module, and custom reflection interference contrast microscopy (RICM) module was driven by Elements software. Dark regions in RICM show where cells have closely adhered to the surface (≪300 nm gap). Nikon 10X objective (CFI60 Plan Fluor) and 60X objective (CFI60 Apochromat TIRF) with DIC filter sets were used. A custom filter (Chroma, zet488/543/638/750m) and lasers (Coherent, 488 and 641 nm; Shanghai Dream Lasers Technology, 543 nm) were used for TIRFM. The long-pass filter (Thorlabs, FGL645) and the custom filter (Chroma, zet488/543/638/750m) were used with the Xenon lamp for RICM. Images were recorded using an electron-multiplying charge-coupled device (EMCCD; Andor iXon 888).

## Measurement of Fab binding and dissociation rate

The flow cell was assembled by sandwiching double-sided sticky tape (Scotch, 665) between a glass slide (Fisher Scientific, 12-544-4) with two holes and a PEGylated coverslip described above. The opening was sealed with epoxy (Devcon, 14250). Tubing (Weicowire, ETT-24N) was connected to the flow cell for stable buffer exchange during the fluorescence measurement. TGT was coated and cells were seeded for 1 h. Fluorescently labeled Fab was injected and washed-out using a syringe pump (Chemyx, Fusion 200) while the fluorescence signal from each spread cell was monitored using TIRFM.

## Image analysis

Image analysis was carried out using ImageJ (v1.53) and custom MATLAB (R2021a) scripts. Close contact enclosed area of cell was analyzed using MATLAB scripts based on image segmentation of RICM images. Cell locomotion tracking and fiducial maker tracking for lateral drift correction was done using ImageJ Trackmate plug-in[70] and MATLAB scripts. For the time-resolved qTGT rupture analysis, time-lapse RICM and TIRFM images and were obtained. The lateral drift was corrected by tracking gold nano particles (Luna Nanotech, GNP-BIOT-100-2-04) in RICM images as fiducial markers. The pixel-wise intensity change was collected frame by frame and median-filtered in time ($N = 3$) to reduce defocusing-induced noise[18].

## Fluorescent signal calibration to measure the single-molecular density

To measure the number of ruptured qTGT or fluorescently labeled Fab, the fluorescent signal intensity of single qTGT or Fab molecules was measured from single-molecule fluorescent signal traces. Photobleaching of the dye on TGT or transient binding of Fab to the surface were collected in regions without cells; the stepwise signal decrease or increase (>40 events) was used for calibration. The imaging for calibration was done on the same surface but away from the area for data acquisition. The calibration was done for every experiment using the equivalent imaging condition except the exposure time.

## Actin flow rate measurement

BJ-5ta cells were seeded on RGD-54pN or LDVP-54pN surfaces for 40 min and 20 nM SiR-actin and 10 μM verapamil (Cytoskeleton, CY-SC001) were added 20 min before imaging. Time-lapse TIRF imaging (640 nm) was done with 2–5 s interval for more than 60 frames. Kymographs were generated with ImageJ and the linear fit analysis was done with MATLAB scripts.

## Traction force microscopy

Polyacrylamide gels (0.69 and 4.07 kPa shear modulus, 40 μm thickness) were prepared on glass coverslips with embedded 40 nm fluorescent beads (Thermo Scientific, TransFluoSpheres 633/720) and the surfaces were coated with neutravidin as described[71]. TGT was immobilized on the surface and the Cy3 signal of TGT was checked to

confirm the immobilization. cRGDfK or MUPA-LDVPAAK ligand was conjugated on the immobilization strand of TGT to prevent force-induced rupture. Cells were seeded on the gels for 1 h. The bead images were taken before and after cell removal with the addition of 0.1% SDS. Bead displacements were determined using particle image velocimetry, and the corresponding contractile energy was estimated with Fourier transform traction cytometry, using ImageJ plugins[72]. The cell area was determined by manually drawing an ROI in the DIC channel. Imaging was performed on Nikon Eclipse Ti2 microscopes equipped with Perfect Focus, CSU-W1 spinning disk, and Hamamatsu Orca-flash 4.0 v3 camera. Illumination was provided by optical fiber (Oz Optics) coupled solid-state lasers: 561 nm (200 mW) for Cy3 and, 655 nm (100 mW) for fluorescent beads from Spectral Applied Research. Emission was collected via single-bandpass emission filter (605/52 nm) and a long-pass LP647 nm filter for far-red from Chroma.

### Cytoskeletal inhibitor experiments
The total number of ruptured qTGT was measured as described above. Cells were seeded after 5 min preincubation with inhibitors. Para-amino-blebbistatin was from Axel (ax494682; 50 uM). Other inhibitors were from Sigma-Aldrich: Y-27632 (Y0503; 10 μM), ML-7 (I2764; 10 μM), Cytochalasin D (C8273; 10 μM), CK666 (SML0006; 50 μM), and SMIFH2 (S4826; 50 μM).

### Immunofluorescence imaging for actin, paxillin, and integrins
Cells were fixed with 4% (w/v) paraformaldehyde in PBS for 10 min and permeabilized with 0.2% (v/v) Triton X-100 for 5 min. Paxillin was sequentially stained with anti-Paxillin antibody (Abcam, ab32084; dilution of 1:200) for 1 h and secondary antibody Alexa Fluor™ 488 goat anti-rabbit IgG (Thermo Fischer Scientific, A-11008; dilution of 1:1000) for 1 h. Actin was stained with SiR-actin (Cytoskeleton, CY-SC001; 10–20 nM) or Alexa Fluor™ 555 Phalloidin (Invitrogen, A34055). Fab was added to the stained cells 10 min before the imaging. Paxillin-TagGFP2 cells were generated using a lentiviral biosensor (Sigma-Aldrich, 17-10154) and sorted by flow cytometry.

### Statistics and reproducibility
Single-cell data were obtained from multiple independent sample preparations for each condition as specified. Percentage of spread cells was obtained from three independent experiments. Statistical analyses were performed using MATLAB software. Two-sided (two sample) t-test was used for two group comparisons. Each group was compared with the control group unless specified otherwise. Representative micrographs are selected from at least two independent experiments.

## Data availability
Source data are provided with this paper. The additional information required to reanalyze the data reported in this paper is available from the corresponding author upon reasonable request. Source data are provided with this paper.

## Material availability
This study generated new synthetic antibodies, IPI-αVβ3.7, IPI-αVβ3.13, IPI-αVβ5.9, IPI- αVβ5.10, IPI-αVβ8.1, IPI-α5β1.2 and IPI-α5β1.4. These antibodies or Fabs were used for immunofluorescent staining or to selectively inhibit specific RGD-binding integrins. The characteristics of these antibodies were reported in supplementary data. These antibodies were generated in collaboration with and are available from The Institute for Protein Innovation, 4 Blackfan Circle, Boston MA 02115. For more information, contact antibodies@proteininnovation.org.

## Code availability
MATLAB scripts for image analysis are available upon reasonable request. Cell area detection and fluorescence analysis scripts are available on GitHub (https://github.com/1molecule/Subtype-Specific_Integrin_Mechanics2022).

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

## Acknowledgements

This work was supported by National Science Foundation grant PHY1430124 (to T.H.) NHLBI grant NIH R01 HL131729 (to T.A.S. and J. L.). T.H. is an investigator of the Howard Hughes Medical Institute. M.H.J. was a recipient of the Korean National Research Foundation grant (2018R1A6A3A03012786) and J.L. was a recipient of Susan G. Komen fellowship (PDF16381021). V.J. and C.M.W. were supported by the Division of Intramural Research of the National Heart, Lung, and Blood Institute. This article is subject to HHMI's Open Access to Publications policy. HHMI lab heads have previously granted a nonexclusive CC BY 4.0 license to the public and a sublicensable license to HHMI in their research articles. Pursuant to those licenses, the author-accepted manuscript of this article can be made freely available under a CC BY 4.0 license immediately upon publication.

## Author contributions

M.H.J., J.L., T.A.S., and T.H. designed research and wrote the paper. M.H.J. carried out TGT based experiments and cell imaging. J.L. measured the integrin expression level and ligand affinity and prepared antibodies and Fabs for imaging, cell staining by flow cytometry and inhibition of RGD-binding integrins. J.L., Y.H., J. C. and J.Y. developed the synthetic antibodies. M.H.J., V.J., C.M.W. designed and conducted traction force microscopy experiment.

## Competing interests

The authors declare no competing interests.
