## [Peer Review File · Nature Communications]

Editorial note: Parts of this Peer Review File have been redacted as indicated where no third party permissions could be obtained.

Reviewers' comments:

Reviewer #1 (Remarks to the Author):

This careful and detailed study employing mechanical measurements using a suite of DNA double helix-based rupturable tension gauge tethers (TGTs). Results show that different integrin classes require and exert different peak forces during spreading and that, depending on the integrin mediating spreading, cells adopt different morphologies associated with different cytoskeletal architectures and actin flows. The manuscript is dense at times but overall it presents a clear, well-described message, the experiments are logical and carefully controlled and I have only very minor suggestions for clarification.

Fig 3A – can counterstaining be used to confirm that the rupture is occurring preferentially at focal adhesions or at other sites of integrin accumulation?

It is not fully clear to me why there is not more rupture at the RGD-TGT 12 and 23? On RGD-TGT 12 and 23 and 33 cells mostly fail to spread and some rupture is seen in Fig 5 but how do the authors envisage that the cell senses the resistance of the tether without pulling it to rupture? Perhaps the authors can explain this more explicitly.

Minor points:

Lines 193-194 – streak patterns should probably refer to fig 3a not 3b

Reviewer #2 (Remarks to the Author):

Comments to „Subtype-specific single beta1 integrin mechanics for activation, mechanotransduction and cytoskeleton remodeling“ by Jo, Li, and colleagues.

In their work, Jo, Li, and colleagues used double-stranded DNA tension gauge tethers (TGTs), of 18bps to measure adhesive forces of cells to their immobilized ECM ligands, and to understand how such mechanical interactions between integrins and their immobilized ligands lead to cellular spreading and signal transduction. Different groups have attempted to understand how-integrin/ligand binding is modulated under mechanical forces, once the integrin/ECM link is established. Previous work with fibronectin fragments or linear RGD peptides surface-coupled via a peptide-spring have shown that integrin/ligand tension are in the range of 2-10pN (ref 9). In their own work (ref 11) the authors have developed a DNA-duplex based force sensor that is coupled to a PEGylated surface, and in which the rupture force is modulated by the differential position of the biotin/surface tether along the oligo-dimer. Strikingly, they have shown that cell adhesion and spreading on a high-affinity, cyclic RGD-analogue occurs only with these constructs in which the calculated rupture threshold (ref 11), corresponds to a tensional force range of 40-50pN (5-10x higher than in other studies, ref 9). Interestingly, this particular observation has been consistently observed with many different cell types, and therefore appears to be a real biological effect. However, the different DNA-duplex constructs show a different geometry in respect to the position of the integrin-binding ligand and the substrate-coupling biotin (see Figure 2a). In the high rupture constructs, the RGD-motif is located on the opposite side of the duplex (6nm in length) in respect to the attachment of the substrate tethering biotin (located on the opposite strand). For the low force sensors, the integrin-binding RGD, is located very close to the substrate tethering biotin, which reduced the spacing from the PEGylated surface, and introduces a different orientation of the highly acidic DNA-duplex.

This different confirmation can be a major problem in this study, as RGD-recognizing integrins are generally in the closed-bent state, they may just not be able to see the presented high-affinity ligand on the PEGylated surface, when bound via the low-force tethers. Accordingly, in figure 3, the authors show essentially no-rupture events for the RGD-tether of the 12pN and 23pN constructs. However, for a high-affinity RGD-ligand, such rupture events should be very frequent, if indeed ligand/integrin interaction is occurring. In fact, a further analysis of the cell-substrate tether strategy shown in figure

8, and evaluation of the occurrence of RGD-ligand rupture events in the substrate-tethered cells should clarify, whether the initial binding reaction between integrins and the tethered RGD-ligands actually occurs. The analysis with the 12G10 antibody is also pointing in this direction (figure 8), proposing that no significant conformational rearrangements of integrins occur on the 12pN RGD-substrate.

Obviously, if this initial integrin/ligand binding event is not occurring, the absence of spreading is trivial, but at the same time reduces the rest of the analysis to a descriptive analysis of the differences between an RGD-high affinity ligand and a linear peptide recognized by a different type of integrin.

If we restrict the analysis only the substrates in which adhesion and spreading occurs, clearly the different ligands produce different phenotypes, and are potentially of interest for a general audience, as it could help to understand the different biological responses of different ECM-substrate, mediated by different classes of integrins. In order to address this issue, the authors use TGTs that are either functionalized with high-affinity cyclic RGD- peptides or a linear LDVP-containing peptide. The use of these specific ligands enabled the authors to study the interaction of different classes of integrins with their respective ligands, specifically (according to the authors) RGD-binding integrins (α V-family and α 5 β 1 integrin) on the one hand, and α 4 β 1 as LDVP binder on the other hand. Several of these integrins are expressed in the cell line that was used in this study (dermal fibroblast), allowing a close comparison of the effects of different ligand/integrin/force-level combinations.

An extensive screening and identification of new synthetic anti-integrin antibodies has allowed the authors to more specifically define the involvement of a given integrin type, especially α v β 1, which can not be analyzed by genetic methods, and this panel of antibodies therefore consists of a high value for the integrin community.

With this experimental system, and the set of integrin-specific antibody reagents, the authors find that cells need a lower force threshold to activate cell spreading when using the LDVP- α 4 β 1 integrin combination compared to RGD-based spreading. They also show that frequent tether rupture events occur with the 12pN construct in the case of the LDVP-construct, but none with the RGD-based construct (Fig 3, see discussion above). Thus the authors claim that the α 4 β 1 integrin-based spreading occurs at a lower tensional threshold than that for RGD-bound integrins (This could potentially be a misinterpretation due to the failure of RGD-recognition on the 12pN constructs, see also ref 9 for spreading on a linear RGD-peptide). Further, by using the specific panels of antibodies, the authors claim that RGD-based adhesion and spreading is dominated by α v β 1 integrin in their RGD-dependent system. In addition, using another cyclic RGD-peptide with a higher specificity and affinity for α 5 β 1 integrin, the authors show that α 5 β 1 integrin is also able to trigger cell spreading in a similar manner to α v β 1 integrin (fig.9). Interestingly, also this construct shows no spreading on low-force tethers, nor rupture of these tethers, requiring also a further analysis of the α 5 β 1 integrin interaction mechanisms during the initial cell substrate binding phase (see discussion above).

Another major problem with the current set of data, is the use of non-physiological integrin ligands for the binding and spreading experiments. Although the use of high-affinity cyclic RGD-containing ligands is helpful to establish a certain ligand/integrin specificity, such cyclic peptide entities are not normally found in physiological proteins. While affinity measures of such artificial ligands are conducted, with either purified integrin receptors, or on cells in suspension (after the addition of Mn^{2+} , which induces the extended-open state of integrins), how such ligands are able to bind to integrins in the low-affinity state, or whether they induce an outside-in conformational activation of the integrin receptor is not completely understood. Especially the latter question could be relevant for the observed phenotypes. When cells are undergoing cell spreading, an important role involves the inside-out activation process of integrins, linked to chemokine signaling in leukocytes, or Rap1-mediated activation of talin. Furthermore, integrin-mediated cell-spreading requires the recruitment of signaling adapters, which trigger the subsequent intracellular signaling steps leading to cell spreading. Interestingly, studies with Mn^{2+} treated melanoma cells have demonstrated talin-mediated α v β 3 integrin clustering on vitronectin substrates, without the respective recruitment of integrin adapters such as vinculin, paxillin or FAK, or even binding to F-actin (Cluzel et al., 2005). In the context of artificial high-affinity integrin ligands, it is therefore possible that integrins get bound to these surfaces, which are not functionally linked to the cytoplasmic adapters to initiate the signaling required for spreading. Here again, it might be highly relevant to understand, whether and which inside-out mechanisms are required to prime the cytoplasmic tail of integrins, in order to transmit the high-affinity ligand mediated cell-substrate

tethering into an intracellular signaling. Or said in other words, if a cell-surface exposed and resting integrin is switched to the extended-open conformation, it may recruit talin, but in a context that is not compatible with cell spreading, or anchorage to the cytoskeleton.

Therefore, it is potentially very important that integrin-dependent substrate recognition, can be performed via low-affinity interactions involving the extended-closed conformation, prior to a force-mediated maturation step. This may give the cell the opportunity and flexibility to form adhesive interactions only in protrusive organelles, such as filopodia and lamellipodia, where the respective integrin adapters are expressed. To gain some insight into such questions, a potential mixing of differentially labeled low and high-affinity ligands might help to better understand the role of low-affinity versus high-affinity integrin conformations during the process of cell adhesion and spreading, as it is well known that different sets of integrin adapter proteins are recruited to these different populations.

Unfortunately, the different topology of the ligands (cyclic versus linear), the different flexibility of the ligand (cyclic/rigid versus linear/flexible), as well as physiological and artificial structures strongly reduces the value of this study for a better understanding of the integrin physiology.

In general, the quality of the presented data is high and the work offers findings that raise interesting questions for the specialist and for people sufficiently familiar with integrin biology. Unfortunately, the interpretation of the data is very superficial, and misleading, especially if readers are not made aware of the non-physiological nature of the integrin ligands and respective experimental settings. For these reasons the study is mainly observational and largely ignores published literature that would have been crucial for an insightful interpretation of the data.

Specifically, while spreading is used as a crucial readout, the authors barely discuss published literature about the interplay of integrins, integrin-mediated signaling including adapters involved in this, and cell spreading. Accordingly, their interpretation of integrin-specific differences in cell spreading remains shallow and only speaks vaguely of connections to the cytoskeleton without offering any mechanistic hypotheses for this interaction. Given that spreading requires intracellular signaling it would be important to discuss differences between $\alpha 4\beta 1$ and RGD-binding integrins when it comes to binding and interaction with cytoplasmic adapters that link these integrins to the actin cytoskeleton and that are involved in cell spreading.

Additionally, the authors stress repeatedly that differences in spreading, cell shape, actin cytoskeleton, etc. between RGD-binding integrins and LDVP-binding integrins are surprising, thereby implying novelty of these findings. At the same time, the authors do cite one publication regarding $\alpha 4\beta 1$ integrin (Kassner et al., 1995, MBoC) that already showed several of the "surprising" effects (reduced spreading, decreased focal adhesion localization, reduced cell adhesive force of $\alpha 4$ compared to $\alpha 2$ or $\alpha 5$ chimera). And in fact, looking at additional published data of the effects on $\alpha 4\beta 1$ integrin on actin cytoskeleton and/or spreading (for example Sechler et al., 2000, JCS; Hight-Warburton et al., 2021, Front Cell Dev Biol), I believe that the observed differences between αV and $\alpha 5\beta 1$ integrins vs. $\alpha 4\beta 1$ integrin are in good alignment with the literature and not that surprising.

A better literature review could have quickly led to an interesting, testable hypothesis: $\alpha 4\beta 1$ integrin has been shown to recruit paxillin via the $\alpha 4$ subunit (refs: Hyduk et al., 2004, Blood; Liu et al., 1999, Nature; Liu and Ginsberg, 2000, JBC). Importantly, paxillin recruitment and signaling is involved in spreading and signaling (refs: Deakin and Turner, 2008, JCS; Pinon et al., 2014, JCB; Theodosiou et al., 2016, eLife), and in cell adhesion forces (refs: Morimatsu et al., 2015, Nano Lett.; Schiller et al., 2011, EMBO Rep). These findings are published several times over many years as shown here and it would have been crucial to put the data in this manuscript into context of this established literature and to reference the people that have worked on these topics. This might have also allowed the authors to add new mechanistic insights to the field beyond the observational data currently presented.

Based on these reasons, I cannot recommend publication in Nature Communications. In order not to mislead readers about established findings vs. novel insights, I would strongly recommend a more thorough literature research and a more insightful discussion irrespective of the journal where this article might be published.

Reviewer #3 (Remarks to the Author):

In the manuscript by Jo et al., the authors developed LDVP-TGT and RGD-TGT, and found that the BJ-5ta fibroblasts adhere, spread and migrate differently. The authors further demonstrated that the different force requirements for spreading are a consequence of the different levels of force exerted by the RGD and LDVP-engaged cytoskeletal machineries. Overall, it is a straightforward investigation with well-designed and carefully conducted experiments. Below are specific suggestions for improvement:

1. A panel of antibodies specific targeting $\alpha V\beta 1$, $\alpha V\beta 3$ or $\alpha V\beta 5$ were developed and used as important tools to identify specific role each integrin subtype plays. The authors described that "we used a different subset of inhibitory IPI and Biogen Fabs with RGD-mimetic heavy chain CDR3 sequences". Please clarify the meanings of "RGD-mimetic sequences". The CDR3s of these antibodies share high homology with the RGD peptide? I suggest to summarize the antibody sequences or at least the CDR3 sequences in a table.

2. The binding affinities (KD) of RGD-mimetic Fabs were calculated against cell surface-expressed integrins. Typically, purified proteins but not cell surface-expressed intergrins should be used to precisely determine the binding kinetics using ELISA or SPR. Also, it is somewhat surprising that all the identified Fabs are inhibitory. Have the authors identified some non-functional binders? Please explain.

Point-by-point response

Before we respond to the comments by the three reviewers point-by-point, let us first address the major methodological concern Reviewer #2 raised (and hinted upon by Reviewer #1). That is, the possibility that the different cell adhesion behaviors between RGD-TGTs of different tension tolerance is due to the difference in ligand accessibility, instead of differences in the rupture forces of the TGTs.

In fact, the issue of TGT geometry along with its potential consequences on ligand accessibility was recognized during the development of TGT technology and was addressed in the original paper of TGT assay as we recap below (Wang and Ha, Science 2013). We understand that such details can be easily missed and have now mentioned them explicitly in the revised manuscript. In addition, many reports from our group and others have shown that TGT-12pN can be recognized and ruptured by various cell types. We further tested it with the cell line used in our manuscript, BJ-5ta cell, and show that RGD-12pN-TGT can be bound and ruptured when there is adequate tension tolerance to support cell spreading. Please see the “Regarding ligand accessibility and rupture of RGD-12pN” below.

For better readability, we use different colors for the below text: **reviewers' comments are in purple**, our answers and explanations are in black, and **manuscript updates are referenced as blue text**.

Regarding ligand accessibility and rupture of RGD-12pN:

The RGD ligand on 12pN-TGT is physically accessible. We have a flexible linker (PEG12 or 6 thymines as described in method section), neutravidin, and another long flexible PEG linker (Biotin-PEG-SVA-5000) between the immobilization strand of TGT and glass substrate. In fact, the ligand accessibility or geometrical difference issue, potentially being responsible for tension threshold for adhesion and spreading, was addressed in the original paper of TGT assay (Wang and Ha, Science 2013). To recap:

Cell spreading was not hindered when the RGD ligand was positioned very near the biotin on single stranded DNA (i.e. non-rupturable) or on the same strand of dsDNA (subfigure A and B), which show that neutravidin or dsDNA extension does not hinder the accessibility of ligand. Highly symmetrical construct versions of TGT-12pN and TGT-56pN were also tested to show that the tension threshold observed for supporting cell spreading is not due to the ligand accessibility difference when coupled to TGTs with different tension thresholds (subfigure C and D).

[REDACTED]

Wang and Ha 2013 (10.1126/science.1231041) Figure S8 legend: “To eliminate the steric hindrance due to
dsDNA or biotin-Avidin that may affect the accessibility of ligands as the source of the observed tension
threshold, we created two control constructs as shown in (A) and (B). Ligand accessibility is expected to be
similar with those of low Ttol RGDfK-DNA constructs shown in fig. S2. However, the tether here is not
rupturable because RGDfK and biotin are on the same DNA strand. Cells adhere normally on both surfaces
showing that ligand accessibility is unlikely to be reason for the observed tension threshold. Two additional
constructs as shown in (C) and (D) were prepared for further tests. The two DNA constructs are highly
symmetrical in overall dimension which should give similar ligand accessibility, but the construct in (C) is in the
unzipping mode and the construct in (D) is in the shear mode. Cells adhered well on the shear mode construct
but not on the unzipping mode construct, indicating that it is not the ligand accessibility but the tension
tolerance causing the difference in cell adhesion.”

**The ligand accessibility of RGD-12pN is also manifested by its ruptures by cells.** Cells do not spread on
RGD-12pN only surface, so the rupture of RGD-12pN is very limited. To show that integrins can bind to the
ligand of TGT-12pN and transmit cellular forces, BJ-5ta cells were seeded on a surface coated with 12pN and
54pN TGTs of different colors (RGD-12pN-BHQ2-Cy3 100nM and RGD-54pN-BHQ2-Atto647N 10nM; the
RGD-54pN TGT of low density facilitates cell spreading on RGD-12pN). In this mixed condition, cells adhered
to the surface and spread and abundant Cy3 signals from ruptured RGD-12pN were observed, showing that
the ligand on 12pN TGT was physically pulled by cells, further supporting high accessibility of ligands.

Our manuscript has been updated, with the results in new supplementary Fig.5:

“Significantly less rupture was observed for RGD-54pN than RGD-43pN (Fig. 3c), showing that much of
the high peak force exerted by RGD-binding integrins during spreading and migration is between 43
and 54 pN. Rupture events of 12, 23 and 33 pN RGD-TGT were relatively fewer because of the much
smaller cell contact area. However, there were significantly more rupture events of RGD-12pN TGT in
the absence than in the presence of competing soluble cRGDfK ligand (100 μ M) (supplementary Fig.
5a). Furthermore, there was significantly more rupture of RGD-23pN than that of RGD-12pN and more
rupture of RGD-33pN TGT than that of RGD-23pN (Fig. 3c). These results demonstrate that peak force
exertion during abortive cell spreading attempts increase in frequency as the tension threshold to
support cell spreading is approached. Abundant RGD-12pN rupture was observed when cell spreading
was facilitated by adding RGD-54pN (supplementary Fig. 5b), indicating that less rupture of RGD-12pN
is not the consequence of low ligand accessibility or geometrical difference compared to TGTs of higher

tension tolerance. This finding is consistent with comparisons of alternative TGT geometries which
showed that differences in cell adhesion on RGD-TGT of different tension tolerance values is not a
result of differences in ligand accessibility or steric hindrance from dsDNA or neutravidin¹¹.”

(New data) **Supplementary Fig. 5. Low force transmission events recorded by qTGT.**

(a) The percentages of cells that produced rupture signals on RGD-12pN. BJ-5ta cells were seeded on RGD-
12pN (BHQ2-Cy3, 100 nM) in the absence or presence of soluble cRGDfK ligand (100 μM). Lines show mean
\pm SE of four independent measurements (total cell number = 36, 34, 35, 50; 44, 26, 25, 16). No cell produced a
significant rupture signal (corresponding to >2000 rupture events after fluorescence signal calibration) in the
presence of soluble cRGDfK. Two sample t-test: *** $p < 0.001$.

(b) BJ-5ta cells were seeded on the surface with two different TGTs (RGD-12pN-BHQ2-Cy3 100 nM and RGD-
54pN-BHQ2-Atto647N 10 nM). Because cells do not spread on RGD-12pN surfaces, a small amount of RGD-
54pN was added to facilitate cell spreading. In this mixed condition, cells adhered to the surface and spread,
but the spreading is limited compared to that of cells on high density RGD-54pN. Abundant Cy3 signals from
ruptured RGD-12pN were observed near the leading edge and filopodia. The calibrated molecular density
maps show that the ligand of RGD-12pN was pulled by cells frequently. The rupture signal of the RGD-54pN
TGT (Atto647N) was negligible in this condition.

We have also previously reported the clear rupture of RGD-12pN using CHO-K1, 3T3 fibroblast and smooth
muscle cells.

[REDACTED]

Jo et al, ACS Biomater. Sci. Eng. 2018 (10.1021/acsbiomaterials.8b01216). Both RGD-12pN and RGD-54pN
rupture signals were monitored in early 3T3 fibroblast spreading on mixed TGT surface (RGD-12pN-BHQ2-
Cy3 500 nM and RGD-54pN-BHQ2-Atto647N 500 nM). Time-resolved signals were shown. In the early stage
of cell spreading, **3T3 fibroblast rupture lots of RGD-12pN. The rupture of RGD-54pN was much less**
**frequently observed in this early stage.** This result was also mentioned in our manuscript (line 488-490)

[REDACTED]

Jo et al, ACS nano 2021 (10.1021/acsnano.1c01782). Cell shortening was induced by histamine for human
airway smooth muscle cells spreading on mixed TGT surfaces (RGD-12pN-BHQ2-Cy3 500 nM and RGD-
54pN-BHQ2-Atto647N 500 nM). The time-resolved signals are shown. **Both RGD-12pN and RGD-54pN**
**rupture were observed in this fast-shortening process.** RGD-12pN rupture signals precede RGD-54pN
rupture signals.

We also list below additional publications from other laboratories that demonstrated robust rupture of RGD-
12pN TGT in various contexts.

RGD-12pN TGT rupture was detected near podosomes (Glazier et al. Nature Communications 2019).
RGD-12pN TGT rupture was detected using a hybridization chain reaction (Duan et al. Angewandte
Chemie 2021).
RGD-12pN TGT rupture was detected with platelets (Yonglinang et al. Biosensors and Bioelectronics
2018)
Super-resolution imaging of ruptured RGD-12pN TGT (Zhao et al. JACS 2020).

In addition, the tension threshold for cell spreading on RGD-TGT has been observed between 33pN and 43pN,
not near 12pN. Furthermore, BJ-5ta cells adhere and spread well on LDVP-12pN which has the same position
of ligand attachment on TGT.

**Answers to the reviewers' comments**

**Reviewer #1 (Remarks to the Author):**

This careful and detailed study employing mechanical measurements using a suite of DNA double helix-based
rupturable tension gauge tethers (TGTs). Results show that different integrin classes require and exert different
peak forces during spreading and that, depending on the integrin mediating spreading, cells adopt different
morphologies associated with different cytoskeletal architectures and actin flows. The manuscript is dense at
93 times but overall it presents a clear, well-described message, the experiments are logical and carefully
controlled and I have only very minor suggestions for clarification.

Thank you for the supportive comments and for the helpful guidance about how to improve our study.

Fig 3A – can counterstaining be used to confirm that the rupture is occurring preferentially at focal adhesions
or at other sites of integrin accumulation?

To address this question, we counterstained paxillin and ruptured qTGTs (LDVP-23pN and RGD-43pN with
BHQ2-Cy3) on live cells and were able to confirm the prediction by the reviewer.

Our manuscript has been updated, with this result in new supplementary Fig.12:

“Integrins αV and $\alpha 4$ did not form clusters on LDVP-54pN and RGD-54pN substrates, respectively (Fig.
7c). Paxillin, a cytoskeletal adaptor protein with the most potential binding partners within focal
adhesions³³, was present on the elongated focal adhesions with αV on the RGD surface and on the
scattered small adhesions with $\alpha 4\beta 1$ on the LDVP surface (Fig. 7d). Paxillin images compared with
ruptured TGTs (LDVP-23pN and RGD-43pN) suggest that force transmission occurred preferentially at
the adhesions (supplementary Fig. 12).”

(New data) **Supplementary Fig. 12. Paxillin recruited near the force transmitting region.**

Paxillin-TagGFP2 lentivirus transfected BJ-5ta cells were seeded on LDVP-23pN (upper panels) or RGD-43pN
(lower panels) with BHQ2 and Cy3 labeled on the ligand strand and immobilization strands, respectively.
TIRFM images of paxillin and ruptured TGT are shown along with RICM images after one hour cell spreading.
Scale bars, 10 μ m.

(Methods) “Paxillin-TagGFP2 cells were generated using a lentiviral biosensor (Sigma-Aldrich, 17–10154) and sorted by flow cytometry.”

It is not fully clear to me why there is not more rupture at the RGD-TGT 12 and 23? On RGD-TGT 12 and 23 and 33 cells mostly fail to spread and some rupture is seen in Fig 5 but how do the authors envisage that the cell senses the resistance of the tether without pulling it to rupture? Perhaps the authors can explain this more explicitly.

The total count of RGD-12/23/33pN rupture is low because most cells fail to initiate spreading on RGD-12/23/33pN (some cells adhere to RGD-33pN). Most cells failed to form close contact to the surface or the close contact area is very small (Fig. 2d). Fig 3c shows the cumulative count of rupture events over one hour. When the cells are seeded on the surface, they will pull and rupture the RGD-12pN or RGD-23pN during the attempts to form adhesions, but the number of ruptures in the small transient contact is much less compared to the cumulative rupture events produced by spreading cells. To clarify, we added an explanation to the revised manuscript (reproduced below). We can also facilitate cell spreading on RGD-12pN by adding RGD-54pN. In this case, much more RGD-12pN TGTs are ruptured. Please also see the section for the ligand accessibility and rupture of RGD-12pN in the beginning of this letter.

Our manuscript has been updated (new supplementary figure):

“Significantly less rupture was observed for RGD-54pN than RGD-43pN (Fig. 3c), showing that much of the high peak force exerted by RGD-binding integrins during spreading and migration is between 43 and 54 pN. Rupture events of 12, 23 and 33 pN RGD-TGT were relatively fewer because of the much smaller cell contact area. However, there were significantly more rupture events of RGD-12pN TGT in the absence than in the presence of competing soluble cRGDfK ligand (100 μ M) (supplementary Fig. 5a). Furthermore, there was significantly more rupture of RGD-23pN than that of RGD-12pN and more rupture of RGD-33pN TGT than that of RGD-23pN (Fig. 3c). These results demonstrate peak force exertion during abortive cell spreading attempts that increase in frequency as the threshold to support cell spreading is approached. Abundant RGD-12pN rupture was observed when cell spreading was facilitated by adding RGD-54pN (supplementary Fig. 5b), indicating that less rupture of RGD-12pN is not the consequence of low ligand accessibility or geometrical difference compared to TGTs of higher tension tolerance. This finding is consistent with comparisons of alternative TGT geometries which showed that differences in cell adhesion on RGD-TGT of different tension tolerance values is not a result of differences in ligand accessibility or steric hindrance from dsDNA or neutravidin¹¹.”

Minor points:

Lines 193-194 – streak patterns should probably refer to fig 3a not 3b

Thank you. Our manuscript has been updated:

“RGD-qTGT rupture showed streak patterns mostly located at the periphery of spread cells (Fig. 3a, b, supplementary Fig. 4b)”

Reviewer #2 (Remarks to the Author):In their work, Jo, Li, and colleagues used double-stranded DNA tension gauge tethers (TGTs), of 18bps to measure adhesive forces of cells to their immobilized ECM ligands, and to understand how such mechanical interactions between integrins and their immobilized ligands lead to cellular spreading and signal transduction.

Different groups have attempted to understand how-integrin/ligand binding is modulated under mechanical forces, once the integrin/ECM link is established. Previous work with fibronectin fragments or linear RGD peptides surface-coupled via a peptide-spring have shown that integrin/ligand tension are in the range of 2–10pN (ref 9). In their own work (ref 11) the authors have developed a DNA-duplex based force sensor that is

167 coupled to a PEGylated surface, and in which the rupture force is modulated by the differential position of the
168 biotin/surface tether along the oligo-dimer. Strikingly, they have shown that cell adhesion and spreading on a
169 high-affinity, cyclicRGD-analogue occurs only with these constructs in which the calculated rupture threshold
(ref 11), corresponds to a tensional force range of 40-50pN (5-10x higher than in other studies, ref 9).
Interestingly, this particular observation has been consistently observed with many different cell types, and
therefore appears to be a real biological effect.

Thanks for summarizing the context of single-molecule integrin force studies. We'd like to explain that TGT
studies are not in conflict with the results of single-molecule FRET peptide spring sensor data from Dunn lab
(ref 9, Chang et al. ACS nano 2016) because these two types of studies measure different forces, thereby
providing complementary information. Please note that our group calibrated the peptide springs used by the
Dunn lab (Grashoff et al, Nature 2010; Brenner et al. Nano Letter 2016), and showed that the FRET-based
tension sensor is sensitive to forces in the range of 3-7 pN. If the force is higher than 7 pN, the FRET efficiency
will be very low (<0.2), and the force magnitude is not quantifiable. This region of force is indicated as ">7pN"
in the histogram shown in Figure 2 from Dunn lab's paper. The authors collected the single-molecule spots
within focal adhesions. As they titled Figure 3b, what they measured is a time-averaged force per sensor
measured in real time. In contrast, TGTs do not detect real time force values, and they rupture only when the
force exceeds their tension tolerance, even for brief moments. High force transmission is relatively rare and
transient, but TGT rupture can record them and show the cumulated events (1 hr cumulation for total rupture
count; 1-2 min cumulation in time-resolved measurements). We agree that most of integrin will be in low
tension region at a given time point (some are not engaged with integrin and others are integrin-engaged but
are under low tension) as reported by the Dunn lab, but TGT studies showed the relatively infrequent high
integrin forces, which will be very difficult to capture using a real time sensor, can play an important role in cell
spreading. The beauty of the TGT approach is that it limits the force transmitted by a single integrin molecule
and keeps the record of high force transmission events even when we are not watching them in real time.

We further clarified this in our manuscript by adding this discussion.

"Forces of a few pN across RGD-binding integrins have indeed been detected using other force
sensors including peptide springs^{9,43,44}. The peptide spring sensors using FRET measure real-time
integrin forces but only in the low force regime (<7 pN)⁹. Forces higher than the detection maximum are
not quantifiable as a result. In contrast, our TGTs detect or modulate integrins transmitting higher force
(12-54pN). High force transmission events would be relatively rare and transient, so most integrins
would be under low tension region at a given time even if they are engaged with the ligand. Because
TGTs can record relatively infrequent high force transmission events, we can quantify the cumulated
number of the events (1 hr cumulation in Fig. 3; 1-2 min cumulation in Fig. 5). Indeed, based on the
steady state number of activated integrins bound to RGD-43pN, we estimated above that integrins
experience high forces only once every seven to twenty minutes, and therefore experienced low forces
at most timepoints. Thus, the results with both types of sensors suggest that at any given time, most
integrins experience low force. The unique contribution of TGT studies is to show that the relatively
infrequent high integrin force events play an important role in regulating cell spreading."

However, the different DNA-duplex constructs show a different geometry in respect to the position of the
integrin-binding ligand and the substrate-coupling biotin (see Figure 2a). In the high rupture constructs, the
RGD-motif is located on the opposite side of the duplex (6nm in length) in respect to the attachment of the
substrate tethering biotin (located on the opposite strand). For the low force sensors, the integrin-binding RGD,
is located very close to the substrate tethering biotin, which reduced the spacing from the PEGylated surface,
and introduces a different orientation of the highly acidic DNA-duplex. This different confirmation can be a
major problem in this study, as RGD-recognizing integrins are generally in the closed-bent state, they may just
not be able to see the presented high-affinity ligand on the PEGylated surface, when bound via the low-force
tethers. Accordingly, in figure 3, the authors show essentially no-rupture events for the RGD-tether of the 12pN

and 23pN constructs. However, for a high-affinity RGD-ligand, such rupture events should be very frequent, if
indeed ligand/integrin interaction is occurring.

The reviewer raised a methodological concern about the ligand accessibility of TGTs with low tension tolerance
here. This issue of TGT geometry along with its potential consequences on ligand accessibility was recognized
during the development of TGT technology and was indeed addressed in the original paper of TGT assay
(Wang and Ha, Science 2013) using a variety of additional DNA constructs (see Figure S8 in that paper,
reproduced in the beginning of this rebuttal letter in the section titled “**Regarding ligand accessibility and**
**rupture of RGD-12pN**”. In that section, we also showed that RGD-12pN is very frequently ruptured by BJ-t5ta
cells when their spreading is facilitated, further supporting high accessibility of ligands.

In fact, a further analysis of the cell-substrate tether strategy shown in figure 8, and evaluation of the
occurrence of RGD-ligand rupture events in the substrate-tethered cells should clarify, whether the initial
binding reaction between integrins and the tethered RGD-ligands actually occurs. The analysis with the 12G10
antibody is also pointing in this direction (figure 8), proposing that no significant conformational rearrangements
of integrins occur on the 12pN RGD-substrate.

Obviously, if this initial integrin/ligand binding event is not occurring, the absence of spreading is trivial, but at
the same time reduces the rest of the analysis to a descriptive analysis of the differences between an RGD-
high affinity ligand and a linear peptide recognized by a different type of integrin.

We believe these comments also arose from the methodological concern about ligand accessibility which we
have addressed above. The reviewer in addition suggests that 12G10 antibody data may be indicative of
ligand inaccessibility. However, this is not the case. Fig. 8e shows a significantly increased 12G10 Fab signal
on 12pN RGD-TGT compared to a control surface, indicating $\beta 1$ integrins are adopting the EO state when they
are engaged with 12 pN RGD-TGT. Please note that the density of 12G10 Fab, not the total Fab signal per
cell, should be compared due to the different cell areas (Fig. 8c).

If we restrict the analysis only the substrates in which adhesion and spreading occurs, clearly the different
ligands produce different phenotypes, and are potentially of interest for a general audience, as it could help to
understand the different biological responses of different ECM-substrate, mediated by different classes of
integrins. In order to address this issue, the authors use TGTs that are either functionalized with high-affinity
cyclic RGD- peptides or a linear LDVP-containing peptide. The use of these specific ligands enabled the
authors to study the interaction of different classes of integrins with their respective ligands, specifically
(according to the authors) RGD-binding integrins (αV -family and $\alpha 5\beta 1$ integrin) on the one hand, and $\alpha 4\beta 1$ as
LDVP binder on the other hand. Several of these integrins are expressed in the cell line that was used in this
study (dermal fibroblast), allowing a close comparison of the effects of different ligand/integrin/force-level
combinations.

An extensive screening and identification of new synthetic anti-integrin antibodies has allowed the authors to
more specifically define the involvement of a given integrin type, especially $\alpha v\beta 1$, which can not be analyzed by
genetic methods, and this panel of antibodies therefore consists of a high value for the integrin community.

The reviewer kindly summarized our effort to dissect the role of each integrin subtype using highly specific
antibodies, many of which were developed by us. We hope that the antibodies we reported will contribute to
future integrin research.

With this experimental system, and the set of integrin-specific antibody reagents, the authors find that cells
need a lower force threshold to activate cell spreading when using the LDVP- $\alpha 4\beta 1$ integrin combination
compared to RGD-based spreading. They also show that frequent tether rupture events occur with the 12pN
construct in the case of the LDVP-construct, but none with the RGD-based construct (Fig 3, see discussion
above). Thus the authors claim that the $\alpha 4\beta 1$ integrin-based spreading occurs at a lower tensional threshold
than that for RGD-bound integrins (This could potentially be a misinterpretation due to the failure of RGD-

recognition on the 12pN constructs, see also ref 9 for spreading on a linear RGD-peptide). Further, by using
the specific panels of antibodies, the authors claim that RGD-based adhesion and spreading is dominated by
α V β 1 integrin in their RGD-dependent system. In addition, using another cyclic RGD-peptide with a higher
specificity and affinity for α 5 β 1 integrin, the authors show that α 5 β 1 integrin is also able to trigger cell spreading
in a similar manner to α V β 1 integrin (fig.9). Interestingly, also this construct shows no spreading on low-force
tethers, nor rupture of these tethers, requiring also a further analysis of the α 5 β 1 integrin interaction
mechanisms during the initial cell substrate binding phase (see discussion above).

The reviewer summarized our main result, the difference tension thresholds for cell spreading mediated by
α 4 β 1, α V β 1 and α 5 β 1. The reviewer repeats the concern that the lack of spreading weak RGD-TGTs is due to
the ligand accessibility. We addressed this issue in the section, "**Regarding ligand accessibility and rupture**
**of RGD-12pN**", in the beginning of this letter (line 1).

Another major problem with the current set of data, is the use of non-physiological integrin ligands for the
binding and spreading experiments. Although the use of high-affinity cyclic RGD-containing ligands is helpful to
establish a certain ligand/integrin specificity, such cyclic peptide entities are not normally found in physiological
proteins. While affinity measures of such artificial ligands are conducted, with either purified integrin receptors,
or on cells in suspension (after the addition of Mn^{2+} , which induces the extended-open state of integrins), how
such ligands are able to bind to integrins in the low-affinity state, or whether they induce an outside-in
conformational activation of the integrin receptor is not completely understood.

Here the reviewer wondered if the peptide ligands we used provide a physiologically relevant setting.
First, fibronectin structures may let us gain insight into why the cyclic forms of the RGD peptide exhibit the high
affinities. In fibronectin, the RGD binding motif is located within a short linker (**GRDGS**) connecting two strands
of a rigid beta-sheet (1FNA and alphafold.ebi.ac.uk/entry/P02751), and as a result, the RGD binding motif
should be highly curved, making the cyclic form an effective mimic. In contrast, LDVP is found within a very
long flexible linker of fibronectin (alphafold.ebi.ac.uk/entry/P02751), thus the linear LDVP motif is an effective
mimic of LDVP in its native context.

To further answer this question, we tested protein ligands. We showed that the characteristic spreading
morphologies on LDVP-TGT and RGD-TGT are reproduced on the protein ligands (VCAM and fibronectin, Fig.
1B). Please note that chemically modified peptide ligands have been widely used in integrin studies. Well
characterized peptide ligands and macromolecule ligands are therefore complementary tool sets for integrin
studies.

To address the questions about the ability of the peptide ligand to bind integrins in the low affinity state
and to induce an outside-in conformational activation of integrins, we measured the integrin affinity of both
versions of RGD peptides, cyclo-RGDfK peptide and cyclic-ACRGDGWCG peptide, under a physiological
condition and the same condition but supplemented with 1mM Mn^{2+} (Fig. 5a, Fig. 9a, Fig. S8, and Fig. S11).

For measuring the two peptides' binding affinity to intact α 5 β 1 and α V β 1 under basal condition (i.e.
without Mn^{2+}), the β 1 subunit extension was monitored using 9EG7 which is a Fab specific to the extended
form of β 1 subunit (Fig. S8a, Fig. S11a). **This 9EG7 epitope exposure detection directly shows that the**
**peptide ligands can induce conformational activation of integrins.** Moreover, for the cyclo-RGDfK peptide,
its binding affinities to intact α 5 β 1 and α V β 1 were also determined when the extended-open (EO)
conformations of β 1 integrin is more populated by using open conformation-stabilizing Fab, 12G10. Under 2
μ M 12G10 Fab, cyclo-RGDfK peptide bind to α 5 β 1 and α V β 1 with 86nM and 132nM affinities, respectively,
which are 7000-fold and 60-fold higher than its basal binding affinity to the respective integrins. For the cyclic-
ACRGDGWCG peptide, we previously showed that its intrinsic binding affinity to the EO conformation of
integrin α 5 β 1 is ~3000-fold higher compared to the closed conformation (Li et al., 2017). All of these indicate
that both versions of RGD peptides do bind to the EO state with much higher affinities and can induce integrin
conformational activation.

These modified ligands are used in the integrin field to establish a certain ligand/integrin specificity, as
pointed out by the reviewer, and also to ensure the adequate affinity to the investigated integrin. In the early
stage of our studies, we only used the cyclo-RGDfK conjugated TGTs, which bind to intact $\alpha V\beta 1$ with 8 μM
affinity, and binds to $\alpha 5\beta 1$ with 610 μM affinity. As we show in Fig.5, even though $\alpha 5\beta 1$ has the highest
expression level among all the RGD-binding integrins expressed on BJ-5ta cell, it could not mediate cell
adhesion and spreading on the cyclo-RGDfK conjugated TGTs due to its low affinity. To directly compare the
mechanical properties of $\alpha 5\beta 1$ and $\alpha V\beta 1$, the two well expressed RGD-binding integrins on BJ-5ta cells
sharing the same $\beta 1$ subunit, we further developed the cyclic-ACRGDWCG peptide conjugated TGT. This
peptide binds to $\alpha 5\beta 1$ and $\alpha V\beta 1$ with comparable binding affinities, 3.5 μM and 6.8 μM , respectively. Either
$\alpha 5\beta 1$ or $\alpha V\beta 1$ can mediate cell spreading on ACRGD-54pN TGT. In comparison, fibronectin binds to $\alpha 5\beta 1$ on
intact cells with an affinity of 1.1 μM (Li et al., 2017), suggesting that ACRGD peptide is a good mimic of
fibronectin in terms of affinity.

Additionally, we point out that affinities for $\alpha 5\beta 1$ of both linear and cyclic RGD peptides and fibronectin
fragments have been measured with soluble $\alpha 5\beta 1$ ectodomains in Mg^{2+} (Li et al., 2017). Large, $\sim 1,000$ fold
increases in $\alpha 5\beta 1$ affinity between closed and open conformations are seen with both fibronectin and RGD
peptides in Mg^{2+} , and saturation binding with both types of ligands stabilizes the high affinity, extended-open
conformation, again showing that cyclic RGD peptide is a good mimic of fibronectin in conformational activation
of integrins. We also demonstrated directly here that binding to cyclo-RGDfK conjugated TGT stabilized the
activated conformation of $\alpha V\beta 1$ as measured with 12G10 Fab (Fig. 8).

As a side note, we recently showed that Mn^{2+} is not sufficient to stabilize the extended-open
conformation (although it partially does so) and one of its major effects is to increase integrin affinity for ligand
independent of conformational state (Anderson et al., MBoC 2022, "Regulation of integrin $\alpha 5\beta 1$ conformational
states and intrinsic affinities by metal ions and the ADMIDAS.").

Especially the latter question could be relevant for the observed phenotypes. When cells are undergoing cell
spreading, an important role involves the inside-out activation process of integrins, linked to chemokine
signaling in leukocytes, or Rap1-mediated activation of talin. Furthermore, integrin-mediated cell-spreading
requires the recruitment of signaling adapters, which trigger the subsequent intracellular signaling steps
leading to cell spreading. Interestingly, studies with Mn^{2+} treated melanoma cells have demonstrated talin-
mediated $\alpha v\beta 3$ integrin clustering on vitronectin substrates, without the respective recruitment of integrin
adapters such as vinculin, paxillin or FAK, or even binding to F-actin (Cluzel et al., 2005). In the context of
artificial high-affinity integrin ligands, it is therefore possible that integrins get bound to these surfaces, which
are not functionally linked to the cytoplasmic adapters to initiate the signaling required for spreading. Here
again, it might be highly relevant to understand, whether and which inside-out mechanisms are required to
prime the cytoplasmic tail of integrins, in order to transmit the high-affinity ligand mediated cell-substrate
tethering into an intracellular signaling. Or said in other words, if a cell-surface exposed and resting integrin is
switched to the extended-open conformation, it may recruit talin, but in a context that is not compatible with cell
spreading, or anchorage to the cytoskeleton.

The reviewer raised the concerns that cells spreading on TGT may lack adhesion signaling and referred to
previous studies which used Mn^{2+} to activate integrins artificially. We did not use Mn^{2+} in our studies here, and
we showed that cells spreading both on RGD-TGT and LDVP-TGT form focal complexes or focal adhesions
enriched with paxillin (Fig. 7d). In addition, cells transmit high forces (12-54 pN) with characteristic patterns:
puncta near leading edge or filopodia and elongating patterns near focal adhesions. Therefore, our data
indicate that various adaptor and signaling proteins are recruited to form dynamic adhesion structures.
Furthermore, we showed the characteristic spreading morphologies on LDVP-TGT and RGD-TGT are
reproduced on the protein ligands (VCAM and fibronectin).

Regarding the observation on Mn^{2+} treated melanoma cells the reviewer mentioned, here is our
thought. Activation of integrins by Mn^{2+} is a benchmark in the integrin field, but how Mn^{2+} works and whether it

reproduces physiological activation is unknown. We recently showed that Mn^{2+} competes with Ca^{2+} at the
ADMIDAS, shifts the conformational equilibrium toward the open state, and increases the affinity of both the
closed and open states of integrin $\alpha 5\beta 1$ by ~30-fold (Anderson et al., MBoC 2022, "Regulation of integrin $\alpha 5\beta 1$
conformational states and intrinsic affinities by metal ions and the ADMIDAS.").

Therefore, it is potentially very important that integrin-dependent substrate recognition, can be performed via
low-affinity interactions involving the extended-closed conformation, prior to a force-mediated maturation step.
This may give the cell the opportunity and flexibility to form adhesive interactions only in protrusive organelles,
such as filopodia and lamellipodia, where the respective integrin adapters are expressed. To gain some insight
into such questions, a potential mixing of differentially labeled low and high-affinity ligands might help to better
understand the role of low-affinity versus high-affinity integrin conformations during the process of cell
adhesion and spreading, as it is well know that different sets of integrin adapter proteins are recruited to these
different populations.

We are not sure what specific concerns that the reviewer is raising that we can address. Under mixed
conditions that the reviewer mentioned, cells spreading would be mainly mediated by the high affinity ligands.
We have tested mixed TGT with two different tension thresholds (one example is shown above) for force
detection and with two different ligands (RGD and LDVP). In the mixed ligand conditions, cells became
elongated as the relative RGD density increased, suggesting that the phenotypes from RGD become
dominant. We are not familiar with the well-known observations on low and affinity integrin conformations cited
by the reviewer.

Unfortunately, the different topology of the ligands (cyclic versus linear), the different flexibility of the ligand
(cyclic/rigid versus linear/flexible), as well as physiological and artificial structures strongly reduces the value of
this study for a better understanding of the integrin physiology.

As peptide ligands have been widely used in the integrin field, we believe well defined peptide ligands are
useful tool sets complementary with physiological protein ligand. Please note that we recapitulated cell
spreading behaviors on peptide conjugated TGTs using biological macromolecule ligands.

In general, the quality of the presented data is high and the work offers findings that raise interesting questions
for the specialist and for people sufficiently familiar with integrin biology. Unfortunately, the interpretation of the
data is very superficial, and misleading, especially if readers are not made aware of the non-physiological
nature of the integrin ligands and respective experimental settings. For these reasons the study is mainly
observational and largely ignores published literature that would have been crucial for an insightful
interpretation of the data.

We hope the reviewer is satisfied by our explanations above concerning ligand accessibility and the ability of
our peptide ligands to recapitulate the abilities of biological protein ligands to conformationally activate integrins
outside-in and to reproduce the adhesion/spreading behavior on biological protein ligand. Using these peptide
ligands that we have validated to be good mimics of protein ligands, we were able to modulate and detect
mechanical forces through single integrin-ligand bonds and examine how these mechanical characteristics
differ between integrin subtypes.

Specifically, while spreading is used as a crucial readout, the authors barely discuss published literature about
the interplay of integrins, integrin-mediated signaling including adapters involved in this, and cell spreading.
Accordingly, their interpretation of integrin-specific differences in cell spreading remains shallow and only
speaks vaguely of connections to the cytoskeleton without offering any mechanistic hypotheses for this
interaction. Given that spreading requires intracellular signaling it would be important to discuss differences

between $\alpha 4\beta 1$ and RGD-binding integrins when it comes to binding and interaction with cytoplasmic adapters
that link these integrins to the actin cytoskeleton and that are involved in cell spreading.

We checked that both integrin subtypes recruit paxillin (different spatial distribution and shape), but we could
not find the difference in composition of adaptor or signaling protein. We found that BJ-5ta cells are difficult to
transfect. Advanced proteomics study may give us clues on the different adhesion signaling, but it is beyond
our competence and scope of this manuscript.

Additionally, the authors stress repeatedly that differences in spreading, cell shape, actin cytoskeleton, etc.
between RGD-binding integrins and LDVP-binding integrins are surprising, thereby implying novelty of these
findings. At the same time, the authors do cite one publication regarding $\alpha 4\beta 1$ integrin (Kassner et al., 1995,
MBoC) that already showed several of the “surprising” effects (reduced spreading, decreased focal adhesion
localization, reduced cell adhesive force of $\alpha 4$ compared to $\alpha 2$ or $\alpha 5$ chimera).

And in fact, looking at additional published data of the effects on $\alpha 4\beta 1$ integrin on actin cytoskeleton and/or
spreading (for example Sechler et al., 2000, JCS; Hight-Warburton et al., 2021, Front Cell Dev Biol), I believe
that the observed differences between αV and $\alpha 5\beta 1$ integrins vs. $\alpha 4\beta 1$ integrin are in good alignment with the
literature and not that surprising.

Thanks for bringing this up. We agree that the observed differences between αV and $\alpha 5\beta 1$ integrins versus
$\alpha 4\beta 1$ are in good alignment with the literature and cite these publications in the revised manuscript. In this
study we add another layer of mechanical understanding to the literature, with a cell line natively co-expressing
these different integrins. The previous literature compared transfected cells, overexpressing only one type of
integrins in each study.

Sechler et al. (2000) studied the role of $\alpha 4\beta 1$ binding to alternatively spliced V region of fibronectin
(containing LDVP motif for $\alpha 4\beta 1$ binding) in fibronectin assembly using $\alpha 4$ transfected CHO(B2) cells, which is
an established cell line deficient in the expression of fibronectin receptors. They found that actin filaments were
largely cortical and do not form stress fibers when the transfectants were cultured on fibronectin, consistent
with our observation of lack of stress fibers on LDVP-TGT. An interesting difference is that our BJ cells do not
migrate on LDVP-TGT whereas the transfected CHO(B2) cells become more migratory.

Hight-Warburton et al. (2021) studied the role of $\alpha 4/\alpha 9$ integrins in epithelial cell migration during wound
healing. They used a keratinocyte cell line natively expressing of $\alpha 4\beta 1$, $\alpha 9\beta 1$ and probably some other
fibronectin binding integrins. They showed one image with scattered $\alpha 4$ integrin clusters in a cell spreading on
fibronectin. The cell shape and elongated focal adhesions implied that the cell spreading was mediated by
RGD-binding integrins, but the punctate $\alpha 4$ clusters look consistent with our study. When inhibiting $\alpha 4\beta 1$ with
Bio1211 or when inhibiting both $\alpha 4\beta 1$ and $\alpha 9\beta 1$ with LDV peptide, they show actin forming more stress fibers
(protrusions) on fibronectin coated surface, which is also consistent with our findings that actin assembled
downstream of $\alpha 4\beta 1$ forms cross-linked networks in circularly spread cells and is in rapid retrograde flow.

We cited Sechler et al.(2000) in the discussion of our revised manuscript:

“We discovered here that in the same cell, the RGD-binding integrins $\alpha V\beta 1$ and $\alpha 5\beta 1$ differ dramatically
from $\alpha 4\beta 1$ integrin in mechanotransduction, i.e. in the types of cytoskeletons they assemble, the forces
those cytoskeletons transmit, and in the way in which these integrins respond to and regulate force
transmission. Integrin $\alpha 4\beta 1$ is expressed in many types of mesenchymal cells, such as fibroblasts, and
also in certain white blood cells such as lymphocytes but not neutrophils. Early studies of $\alpha 4$ subunit
chimeras examined their function in K562 and CHO cell transfectants. Compared to chimeras
containing $\alpha 4$ cytoplasmic domains, chimeras lacking cytoplasmic domains and those with $\alpha 5$ and $\alpha 2$
cytoplasmic domains favored firm adhesion over rolling adhesion, greater spreading, greater
association with focal adhesions, and lesser cell migration³⁶. However, actin cytoskeleton structure was
not examined in this study. Another study compared $\alpha 4$ and $\alpha 5$ -transfected cells and showed formation

of cortical actin and more migration with $\alpha 4$ compared to actin in stress fibers with $\alpha 5$ ³⁷. Chimeras with
$\alpha 11b$ similarly showed that the $\alpha 4$ cytoplasmic domain antagonized spreading and that distinct biological
responses were due to binding of paxillin to $\alpha 4$ ³⁸, which required binding to cytoplasmic domain
residues Glu-983 and Tyr-991 and was blocked by phosphorylation at intervening residue Ser-988^{39,40}.
Compared to previous reports, we show distinctive RGD-binding and $\alpha 4$ integrin, actin and paxillin
cytoskeleton architectures, rounder cells with $\alpha 4$ integrins, that $\alpha 4\beta 1$ integrins in BJ-5ta cells did not
mediate cell migration, and that the threshold tension required for spreading and the magnitude of force
exertion by RGD-binding and $\alpha 4$ integrins differ.”

A better literature review could have quickly led to an interesting, testable hypothesis: $\alpha 4\beta 1$ integrin has been
shown to recruit paxillin via the $\alpha 4$ subunit (refs: Hyduk et al., 2004, Blood; Liu et al., 1999, Nature; Liu and
Ginsberg, 2000, JBC). Importantly, paxillin recruitment and signaling is involved in spreading and signaling
(refs: Deakin and Turner, 2008, JCS; Pinon et al., 2014, JCB; Theodosiou et al., 2016, eLife), and in cell
adhesion forces (refs: Morimatsu et al., 2015, Nano Lett.; Schiller et al., 2011, EMBO Rep). These findings are
published several times over many years as shown here and it would have been crucial to put the data in this
manuscript into context of this established literature and to reference the people that have worked on these
topics. This might have also allowed the authors to add new mechanistic insights to the field beyond the
observational data currently presented.

Thanks for the suggestion. We added the following paragraph on paxillin to Discussion:

“Paxillin is a multi-domain scaffold recruiting numerous regulatory and structural proteins that together
control the dynamic changes in cell adhesion and cytoskeletal reorganization⁴⁷. Quantitative mass
spectrometry for integrin adhesion complexes suggested that LIM domain proteins, including paxillin,
are potential tension sensors⁴⁸. In fibroblasts, paxillin shows high degree of spatial correlation with
RGD-binding integrin force⁴⁴ and directly binds to kindlin, which is crucial to activate $\beta 1$ integrins and
adhere to fibronectin⁴⁹. Even though we observed that paxillin was recruited both in $\alpha 4\beta 1$ and RGD-
binding integrin mediated adhesions, the difference in mechanosensing and force transmission implies
that the paxillin signaling may work in different manners. Unlike $\alpha 11b$, $\alpha 5$ and $\beta 1$ tails, the $\alpha 4$ tail binds
tightly to paxillin, which regulates cell spreading^{38,40}. This direct paxillin interaction may underlie the low
tension threshold for $\alpha 4\beta 1$ -mediated cell spreading and its distinctive cytoskeleton architecture.”

Based on these reasons, I cannot recommend publication in Nature Communications. In order not to mislead
readers about established findings vs. novel insights, I would strongly recommend a more thorough literature
research and a more insightful discussion irrespective of the journal where this article might be published.

We have extensively revised the manuscript, adding new data, explicitly addressing the issues of ligand
accessibility and physiological relevance of peptide ligands used, and provide an expanded survey of the
literature to put our findings in a broader context.

Reviewer #3 (Remarks to the Author):

In the manuscript by Jo et al., the authors developed LDVP-TGT and RGD-TGT, and found that the BJ-5ta
fibroblasts adhere, spread and migrate differently. The authors further demonstrated that the different force
requirements for spreading are a consequence of the different levels of force exerted by the RGD and LDVP-
engaged cytoskeletal machineries. Overall, it is a straightforward investigation with well-designed and carefully
conducted experiments. Below are specific suggestions for improvement:

1. A panel of antibodies specific targeting $\alpha V\beta 1$, $\alpha V\beta 3$ or $\alpha V\beta 5$ were developed and used as important tools to
identify specific role each integrin subtype plays. The authors described that “we used a different subset of

inhibitory IPI and Biogen Fabs with RGD-mimetic heavy chain CDR3 sequences". Please clarify the meanings of "RGD-mimetic sequences". The CDR3s of these antibodies share high homology with the RGD peptide? I suggest to summarize the antibody sequences or at least the CDR3 sequences in a table.

"RGD-mimetic sequences" means that CDR3 of the heavy chain variable region contains RxD motif. We have added a supplementary table 2 summarizing the heavy-chain CDR3 sequences of the synthetic antibodies used in this study.

	Heavy-chain CDR3 sequences
IPI- α 5 β 1.2	APGGSVYG
IPI- α 5 β 1.4	QRGLLRPAYG
IPI- α V β 3.7	RVSNSAR RGD VRVGY
IPI- α V β 3.13	REHIAG RLD DVYYY
IPI- α V β 5.9	AFVRW RGD SLVLSTW
IPI- α V β 5.10	FLGFGRY
Biogen- α V β 1.5	GGPT RGD GTRVYYYGMDV
Biogen- α V β 1.9	GLWSTEVRYYYMDV
Biogen- α V β 1.10	GLWSTEVRYYYMDV

Supplementary Table 1. Heavy-chain CDR3 sequences of the synthetic antibodies

Heavy-chain CDR3 sequences of the synthetic antibodies used in this study. RxD motif is highlighted.

2. The binding affinities (KD) of RGD-mimetic Fabs were calculated against cell surface-expressed integrins. Typically, purified proteins but not cell surface-expressed integrins should be used to precisely determine the binding kinetics using ELISA or SPR. Also, it is somewhat surprising that all the identified Fabs are inhibitory. Have the authors identified some non-functional binders? Please explain.

Thank you for bringing this up. It is more appropriate to measure affinities for cell surface integrins, as ligand affinity depends on the conformational state of the integrin, and integrins lacking transmembrane domains have much higher affinity as a result of higher population of the high affinity state. The only way that we can accurately determine the concentration required for inhibition of cell function is to do the assays on intact cells. On the other hand, many labs use commercially sourced, uncharacterized integrins, and ELISA readouts are non-equilibrium. Our assays are done under equilibrium, reversible conditions, are highly accurate, are based on many previous studies, show good fits to binding isotherms, and are shown in the supplement. Since we used two of these antibodies to specifically inhibit cell surface expressed integrin α V β 3 or α V β 5, we measured their binding affinities to the target integrins, as well as the other RGD-binding integrins expressed on BJ-5ta cell surface. In this way, we can calculate the Fab concentration to selectively inhibit desired integrins.

Many non-inhibitory antibodies were obtained as well. The prevalence of inhibitory Fabs in our screening outcome may be related to the methodology. In the first magnetic-activated cell sorting, 6 RGD-binding integrins, α V β 1, α V β 3, α V β 5, α V β 6, α V β 8 and α 5 β 1 were used together in the screening, thus a lot of Fabs with RGD-mimetic heavy chain CDR3 may have been selected. In the subsequent FACS sorting, yeast were subjected to positive selection with target integrin ectodomain, and negative selection against all the untargeted integrins.

We have updated our manuscript to clarify this:

"For inhibition studies we used a different subset of inhibitory IPI and Biogen Fabs with RGD-mimetic heavy chain CDR3 sequences (supplementary Table 1). Since we used these antibodies to specifically inhibit cell surface expressed integrins, we measured their binding affinities to the target integrins, as well as the other RGD-binding integrins expressed on BJ-5ta cell surface (Fig. 6c and supplementary Fig. 10a). They show selectivities for α V β 1, α V β 3 and α V β 5 ranging from 160 fold to 2,000 fold."

REVIEWER COMMENTS

Reviewer #1 (Remarks to the Author):

The authors have addressed my concerns.

Reviewer #2 (Remarks to the Author):

This reviewer would like to thank the authors for having clarified a number of issues that have been raised during the first revision. Unfortunately, some of the relevant information to better understand the experimental set-up are hard to find, and it is therefore very important that the authors inform the readers about these critical data in the introduction of the manuscript. This limited information principally concerns a point also raised by reviewer one, and which I did not find properly addressed in the rebuttal letter. What happens to the RGD-ligands on low-tether force substrates during the cell spreading? Or, alternatively, what is happening with RGD-bound integrins, when activated on surfaces with low-tether forces? Do these activated integrin/ligand complexes get internalized, or do they even generate intracellular signaling that is relevant for the spreading process?

In Wang et al (2015) *Biophys J* 109,2259-2267, the authors actually have shown that low-force tethered RGD is extensively cleared from the surface underneath the "non-spread" cell, and subsequently internalized. Blocking of the intracellular contractility with blebbistatin, affects not the "clearing" of the low-force tethered RGD, but reduces the rupture of the high-force tethered RGD. Thus the data from the authors are not new in this respect, and should be clearly introduced and explained. Independently of the cell contraction issue, the observation that integrin/ligand complexes are internalized is extremely important for the integrin-biology, as it may explain, why cells cannot spread on low-force tethered RGD-ligands. Since integrins are bound to non-physiological high-affinity ligands, it is not clear whether such internalized integrins are recycled, or alternatively degraded by the cells (Lobert et al (2010), *Dev Cell* 19, 148-159). Since Focal adhesion kinase can be detected associated with activated and ligand-bound integrins in internalized vesicles (Alanko et al (2015), *Nat Cell Biol* 17 1412-21), or (Fuentes et al (2020) *Nat Comm* 11 4261) a significant intracellular signaling is induced by the low force tethered ligands that is likely to negatively affect the capacity of the cells to spread on this surface. The cell spreading experiment performed on a mixture of low- and high-force tethers (Fig 1 in rebuttal letter) actually confirms the extensive internalization of the low-force tethered RGD at sites of adhesion by the high-force tethered RGD's, while negatively affecting the overall cell spreading capacity of the cell. Therefore additional titration experiments are required to better assess the negative effect of internalized and ligand-bound integrins on the overall cell spreading capacity of cells.

The use of non-physiological high-affinity RGD-ligands could therefore be a problem for the spreading of the cells on the low-force tethers, because they cannot efficiently remove the ligand from their integrins without interfering with their own capacity for spreading. In addition, when such internalized integrin/ligand complexes are activating intracellular FAK, which reportedly increases RhoA activity, the cell will not be able to adapt and prevent ligand rupture by reducing their contractility on low-force tethered surfaces.

For the introduction, it is therefore important to mention that the internalization of soluble (or easily removed) high affinity ligands causes an agonistic effect and integrin-mediated signaling that could negatively effect cell spreading. In turn, it would be important to know whether FAK, or src inhibitors would allow to overcome this negative signaling effect, in order to maintain cell spreading on low-force tethers as it is the case for $\alpha 4\beta 1$ ligands.

In addition to the introduction of the integrin literature, it would also be relevant to mention papers putting the cell spreading in context to membrane-tension and focal adhesion formation (Gauthier et al (2011), *PNAS* 108, 14467; Pontes et al (2017) *JCB* 216 2959). These papers show that cell spreading occurs in two phases, involving a first phase of excretion of membrane from internal stores which is followed by a second phase in which cell edge extension is linked to cycles of extension and retraction which positions new focal adhesions at the cell periphery. The partial spreading on $\alpha 4\beta 1$ ligands on the low-force tethers, and the subsequent increase in cell surface area on the high-force tethers would fit this observation, and the two phases of spreading shown in the current data. Furthermore it would be complementing the previous observations by the authors (Wang and Ha, 2013) that lowering membrane tension, can lead to cell spreading on RGD-33pN tethers, showing that the modulation of cell spreading is coupled not only to myosin-dependent but also plasma membrane

tension. This information is important so that readers can evaluate the differences observed in the current manuscript in which spreading is analyzed on different ligands.

While the discussion has been improved to include alternative mechanisms that could explain the differential spreading on RGD versus LDV ligands, involving mainly intracellular adapters. The authors have not exhausted their possibilities to understand the differential cell spreading at the level of the integrin binding site. Both RGD-ligands used in this study are based on cyclic constructs that will only slightly change their geometry when put under mechanical tension. This situation is completely different for the LDV-peptides, which represent a linear arrangement of the integrin ligand. In the linear arrangement, increasing tether forces are directly affecting the position of the amino acids binding to the integrin pocket, an effect that would be much reduced when under pull, the RGD-loop is maintained. I acknowledge that the authors are using RGD-containing ligands, or VCAM, and are able to reproduce the differential cell spreading behavior. However importantly, this comparison was done under high tether forces, and force-threshold relevant for the morphological changes cannot be evaluated by plating cells on high-force ligands only. To clearly make the point, and to better understand how integrin ligands change their behavior, between a diffusion controlled interaction with integrins, or when put under mechanical tension, a linear RGD-peptide should be used for the experiments as well. Such linear peptides are found in osteopontin or vitronectin and it is actually not known, whether force thresholds would change if the RGD-ligand would undergo conformational changes in respect to mechanical tension. If integrins would just slip-off from linear RGD-peptides, we could assume that similar to the behavior on LDV-ligands, cells can adapt to the lower force thresholds of the ligand-substrate tether, allowing spreading also under low-force tether conditions. I can understand that this is a major experimental effort, but since this major structural difference between the RGD- and LDV-ligands is not further explored, the manuscript remains descriptive and at the moment reproduces well known data from the literature with a new experimental system that is unfortunately insufficiently characterized in respect to the fate and downstream signaling capacities of the internalized (low-force tethered) ligands.

Reviewer #3 (Remarks to the Author):

The authors have addressed my concerns.

<Point-by-point response>

Our answer to Reviewer #2 is colored in blue.

Reviewer #2 (Remarks to the Author):

This reviewer would like to thank the authors for having clarified a number of issues that have been raised during the first revision. Unfortunately, some of the relevant information to better understand the experimental set-up are hard to find, and it is therefore very important that the authors inform the readers about these critical data in the introduction of the manuscript. This limited information principally concerns a point also raised by reviewer one, and which I did not find properly addressed in the rebuttal letter.

We note that Reviewer 1 was satisfied with our responses and revision in response to their comment that Reviewer 2 refers to.

“The rupture events of 12, 23 and 33 pN RGD-TGT are relatively fewer because cell area is much smaller. Interestingly, there was also significant rupture of RGD-23pN and RGD-33pN TGT, showing peak force exertion during abortive cell spreading attempts. Abundant RGD-12pN rupture was observed when cell spreading was facilitated by adding RGD-54pN (supplementary Fig. 5), which suggests that less rupture of RGD-12pN is not the consequence of low ligand accessibility or geometrical difference compared to higher tension thresholds.”

What happens to the RGD-ligands on low-tether force substrates during the cell spreading? Or, alternatively, what is happening with RGD-bound integrins, when activated on surfaces with low-tether forces? Do these activated integrin/ligand complexes get internalized, or do they even generate intracellular signaling that is relevant for the spreading process?

On weak tether surfaces, we observed minimal integrin/ligand internalization as we will show to Reviewer 2 below. Although some integrins are conformational activated on such a surface, they do not rupture enough TGTs for detecting internalization the assay from Wang et al paper in Biophysical Journal 2015 that Reviewer 2 referred to and they do not lead to cell spreading.

In Wang et al (2015) Biophys J 109,2259-2267, the authors actually have shown that low- force tethered RGD is extensively cleared from the surface underneath the “non-spread” cell, and subsequently internalized. Blocking of the intracellular contractility with blebbistatin, affects not the “clearing” of the low-force tethered RGD, but reduces the rupture of the high-force tethered RGD. Thus the data from the authors are not new in this respect, and should be clearly introduced and explained.

We updated the text as shown below in response to this request.

“RGD-54pN was significantly suppressed by each of these three inhibitors (Fig. 3d), indicating that actomyosin contraction exerts forces above 54 pN through RGD-binding integrins, as previously shown for CHO-K1 and 3T3 cells^{19,27}.”

Independently of the cell contraction issue, the observation that integrin/ligand complexes are internalized is extremely important for the integrin-biology, as it may explain, why cells cannot spread on low-force tethered RGD-ligands. Since integrins are bound to non-physiological high-affinity ligands, it is not clear whether such

internalized integrins are recycled, or alternatively degraded by the cells (Lobert et al (2010), Dev Cell 19, 148-159). Since Focal adhesion kinase can be detected associated with activated and ligand-bound integrins in internalized vesicles (Alanko et al (2015), Nat Cell Biol 17 1412-21), or (Fuentes et al (2020) Nat Comm 11 4261) a significant intracellular signaling is induced by the low force tethered ligands that is likely to negatively affect the capacity of the cells to spread on this surface.

Although ligand internalization is an interesting topic that merits its own study in the future, we do not believe this is relevant to our current study because we did not detect ligand internalization using the same approach Wang et al used in that 2015 Biophysical Journal paper (Figure R1). This is not surprising because unlike in CHO-K1 cells used for Wang et al, which extensively ruptures weak TGTs, BJ-5ta fibroblasts do not rupture weak TGTs extensively (Figure R2 and R3). We also separately confirmed that CHO-K1 cells rupture weak TGTs extensively (Figure R4). Therefore, any potential TGT internalization effect on cell spreading should be minimal.

Figure R1 | BJ-5ta fibroblast on RGD-12pN (Cy3 on ligand-strand, 30 min). DIC (top) and Cy3 (bottom) images at six different heights ($h = 0$ on the substrate). 60X objective was used. Scale bar, 10 μm .

Figure R2 | A representative BJ-5ta fibroblast on RGD-q-12pN (1 hr; BHQ2-Cy3). DIC (left), RICM (middle), and TIRFM image of ruptured RGD-q-12pN (right; very low signal). Scale bar, 10 μm .

Figure R3 | RGD-q-12pN and LDVP-q-12pN ruptured by BJ-5ta fibroblast for 1 hr. A box and whisker plots indicate percentiles (5, 25, 50, 75, and 95). $N = 106$ and 104 cells. The same data set is shown with main Figure 3c.

Figure R4 | A representative CHO-K1 cell on RGD-q-33pN (1 hr). DIC (left), RICM (middle), and TIRFM image of ruptured RGD-q-33pN (right). Scale bar, 10 μ m.

The cell spreading experiment performed on a mixture of low- and high-force tethers (Fig 1 in rebuttal letter) actually confirms the extensive internalization of the low-force tethered RGD at sites of adhesion by the high-force tethered RGD's, while negatively affecting the overall cell spreading capacity of the cell. Therefore additional titration experiments are required to better assess the negative effect of internalized and ligand-bound integrins on the overall cell spreading capacity of cells.

As explained above, internalization of ligand bound integrins is minimal in our system.

The use of non-physiological high-affinity RGD-ligands could therefore be a problem for the spreading of the cells on the low-force tethers, because they cannot efficiently remove the ligand from their integrins without interfering with their own capacity for spreading.

Actually, the integrin affinities of RGD ligands are comparable to that of fibronectin as presented in our supplementary figures already. Below, for Reviewer 2's convenience, we created a summary table of dissociation constants of RGD ligands used in this manuscript.

Ligand	$\alpha 5\beta 1$ full length	$\alpha V\beta 1$ full length	Data location
Fn3 9-10	930 nM	10000 nM	Fig. S9a
cRGDfK	610000 nM	8000 nM	Fig. S9a
ACRGDGWCG	3680 nM	6800 nM	Fig. S13a

In addition, when such internalized integrin/ligand complexes are activating intracellular FAK, which reportedly increases RhoA activity, the cell will not be able to adapt and prevent ligand rupture by reducing their contractility on low-force tethered surfaces. For the introduction, it is therefore important to mention that the internalization of soluble (or easily removed) high affinity ligands causes an agonistic effect and integrin-mediated signaling that could negatively effect cell spreading. In turn, it would be important to know whether FAK, or src inhibitors would allow to overcome this negative signaling effect, in order to maintain cell spreading on low-force tethers as it is the case for $\alpha 4\beta 1$ ligands.

Because BJ-5ta, which does not spread on RGD-12pN, does not extensively rupture the weak TGTs or internalize them (see above), we believe internalization and its impact on signaling is not relevant to our study. However, we agree with the reviewer that integrin/ligand internalization and associated signaling is a fascinating topic worthwhile an in-depth investigation for cell types that show extensive ligand/receptor internalization at low forces.

In addition to the introduction of the integrin literature, it would also be relevant to mention papers putting the cell spreading in context to membrane-tension and focal adhesion formation (Gauthier et al (2011), PNAS 108, 14467; Pontes et al (2017) JCB 216 2959). These papers show that cell spreading occurs in two phases,

involving a first phase of excretion of membrane from internal stores which is followed by a second phase in which cell edge extension is linked to cycles of extension and retraction which positions new focal adhesions at the cell periphery. The partial spreading on $\alpha 4\beta 1$ ligands on the low-force tethers, and the subsequent increase in cell surface area on the high-force tethers would fit this observation, and the two phases of spreading shown in the current data. Furthermore it would be complementing the previous observations by the authors (Wang and Ha, 2013) that lowering membrane tension, can lead to cell spreading on RGD-33pN tethers, showing that the modulation of cell spreading is coupled not only to myosin-dependent but also plasma membrane tension. This information is important so that readers can evaluate the differences observed in the current manuscript in which spreading is analyzed on different ligands.

Thanks for the suggestion of more references. Gauthier et al (2011) showed that fibroblast spread rapidly and isotropically, in the first (early) phase, with low level of membrane excretion (exocytosis) and contraction. Pontes et al (2017) observed how changes in membrane tension influence the actin cytoskeleton and adhesion behaviors in the lamellipodium. Wang and Ha (2013) showed that increasing membrane tension, using a hypotonic condition, can lower the tension threshold for cell spreading (~10% increase in the percentage of adhered CHO-K1 cell on RGD-33pN). To our knowledge, some cells show protrusion-contraction whereas others do not show contraction. And the relationship between membrane tension and our study is not clear yet. Because membrane tension is not a subject that our study directly addressed, we do not believe we need to add these references and discuss them separately.

While the discussion has been improved to include alternative mechanisms that could explain the differential spreading on RGD versus LDV ligands, involving mainly intracellular adapters. The authors have not exhausted their possibilities to understand the differential cell spreading at the level of the integrin binding site. Both RGD-ligands used in this study are based on cyclic constructs that will only slightly change their geometry when put under mechanical tension. This situation is completely different for the LDV-peptides, which represent a linear arrangement of the integrin ligand. In the linear arrangement, increasing tether forces are directly affecting the position of the amino acids binding to the integrin pocket, an effect that would be much reduced when under pull, the RGD-loop is maintained. I acknowledge that the authors are using RGD-containing ligands, or VCAM, and are able to reproduce the differential cell spreading behavior. However importantly, this comparison was done under high tether forces, and force-threshold relevant for the morphological changes cannot be evaluated by plating cells on high-force ligands only. To clearly make the point, and to better understand how integrin ligands change their behavior, between a diffusion controlled interaction with integrins, or when put under mechanical tension, a linear RGD-peptide should be used for the experiments as well. Such linear peptides are found in osteopontin or vitronectin and it is actually not known, whether force thresholds would change if the RGD-ligand would undergo conformational changes in respect to mechanical tension. If integrins would just slip-off from linear RGD-peptides, we could assume that similar to the behavior on LDV-ligands, cells can adapt to the lower force thresholds of the ligand-substrate tether, allowing spreading also under low-force tether conditions.

I can understand that this is a major experimental effort, but since this major structural difference between the RGD- and LDV-ligands is not further explored, the manuscript remains descriptive and at the moment reproduces well known data from the literature with a new experimental system that is unfortunately insufficiently characterized in respect to the fate and downstream signaling capacities of the internalized (low-force tethered) ligands.

To address this point, we performed an experiment using a linear RGD-TGT (Figure R5). The linear peptide sequence is GRRGDLATIHGGK, and we used GRRGD-TGT as an abbreviation. As shown in Fig. R5, the GRRGD-TGT showed no spreading for weak TGTs, with tension tolerance below 40 pN, and showed spreading for TGTs with tension tolerance above 40 pN. Therefore, the 40 pN threshold we observed using the cyclized

RGT-TGTs is reproduced using the linear RGD, showing that the high tension threshold for cell spreading is not an artifact of ligand cyclization.

Figure R5 | BJ-5ta fibroblasts on linear RGD ligand surfaces. Cells were seeded for one hour on TGTs conjugated with a linear RGD peptide (GRRGDLATIHGGK; GRRGD-TGT). GRRGD-TGT (100 nM) was immobilized on PEG-passivated and neutravidin functionalized surface. GRRGD-54pN was incubated without neutravidin for control. Images (left) and cell area measured from RICM images (right) are shown. Scale bar, 10 μm.

REVIEWERS' COMMENTS

Reviewer #2 (Remarks to the Author):

Review to Jo et al.

I would like to thank the authors for their efforts to explain the experimental results in detail and for their efforts to try even another linear RGD-peptide for their binding studies.

My main problem to understand the presented data, was to follow the fate of the activated and ligand-bound integrins on the 12pN-RGD-tether. If these ligands are strongly bound to ligands, and are able to couple to the retrogradely moving actin cytoskeleton, we should be able to observe extensive rupture of the low-force tethers. This rupture, however, is not occurring and explanations for this effect are still missing. One solution to this problem could be that despite the ligand induced out-side in activation, the cell is not able to couple these integrins to the rearward moving actin cytoskeleton. I therefore asked the authors to put cells on substrates simultaneously presenting low and high-force RGD tethers. As nicely shown by the authors, cell shape, and frequency of tether ruptures are completely different to the conditions with individual force tethers. Cells do not fully spread on such mixtures, and show extensive rupture of the 12pN-RGD-ligands, occurring at the leading edge of the spread cell, very similar to what is shown on the low-force tethered LDV-ligands. That spreading is not complete is an indication that the rupturing of low-force tethers is inducing a retro-control on the actin cytoskeleton, controlling cell spreading also on softer surfaces. In addition, this experiment clearly shows that 12pN-RGD-tethers can be recognized by RGD-integrin and are pulled from the substrate, when seen in the context of the leading edge of the lamellipodium, but apparently not during the initial phases of cell-substrate interactions via filopodia.

In another key experiment, the authors attach the cells to the substrate, via a non-integrin-dependent cholesterol-linker, and observed significant activation (EO conformation) of the RGD-dependent integrins, without inducing spreading, nor significant rupture of the 12-pN tether. Only when the tether force-resistance is above 33-pN, cells also begin to spread on these surfaces. As the authors correctly state, the activation state of the integrins is not coupled to the signaling/spreading activity, nor tether rupture events.

While the authors put a big emphasis on the dissection of the extracellular mechanisms, the key difference between $\alpha 4\beta 1$ and $\alpha v\beta 1$ (or $\alpha 5\beta 1$) is likely to be situated at the interface between the integrin cytoplasmic tails and the retrogradely moving actin filaments. As nicely shown, perturbation of the lamellipodial F-actin flow (by the formin inhibitor), but not myosin-activity, blocks the spreading/rupture events. This puts the finger on the integrin/F-actin link and the potential reason for the observed differences. Clearly one possible explanation of the differences between $\alpha 4\beta 1$ and $\alpha 5\beta 1$ integrins are the observation that the $\alpha 4$ -cytoplasmic tail is special (several references), having notably the capacity to recruit paxillin directly. Possibly, direct paxillin recruitment to integrins could induce integrin-mediated signaling that otherwise needs to be provided by the tension-dependent recruitment of paxillin to the integrin/talin/kindlin complex. Relatively recent data, shows that paxillin is not only a signal transducer, but is directly involved in the physical maturation of the integrin/talin/actin linkage, and can when directly recruited to the integrin receptor induce focal adhesion maturation (see Ripamonti et al., 2021). The leukocyte expressed $\alpha 4\beta 1$ integrin is also special concerning its requirement for RIAM-mediated talin activation. While $\beta 2$ -integrin (LFA-1) activation requires Rac1/RIAM, RIAM is not required for $\alpha 4\beta 1$ integrin-dependent leukocyte adhesion. Although $\beta 2$ -integrins are not discussed in this manuscript, they are known for their association with filamin, which binds similar to myosin-X to integrins prior to their activation via an inside-out activation mechanisms. Filamin association with $\beta 1$ integrins outside of focal adhesions has been shown by fluorescence correlation microscopy by the Horwitz group.

Here in this manuscript, the authors show that high-affinity integrin ligands induce conformational activation of the extracellular integrin domains, well below the 12pN-force threshold. However, it is not clear when and how this conformational change is actually transmitted to the integrin cytoplasmic tails. What is happening to integrin extracellular domain, when filamin is providing the mechanical interaction to the retrogradely moving F-actin filaments? Will the integrin dissociate from the high-affinity ligand below the 12-pN-force threshold? What happens with the integrin when the linkage to the F-actin is not reinforced by the talin/kindlin linkage? Potentially, we would expect to find the integrin pulled out of the cell membrane, even by the relatively low force 12-pN tether. Pulling integrins out of the cells has been previously observed, when migrating cells are retracting from high

affinity ligands, or seen in experiments in drosophila embryos, during body wall muscle contraction especially in response to a weakening of the integrin/talin linkage.

These are potential aspects the authors could address in further studies, however after several revisions, the study is now sufficiently well referenced and shows the critical experimental details and data, from which the informed reader should be able to draw similar conclusions, to the once, I just mentioned.

Minor issues:

Line 142 : the statement that plasma fibronectin is not expressing the $\alpha 4\beta 1$ binding V-segment is not correct: Plasma fibronectin does not express the EIIIA and EIIIB domains, but the fibronectin dimer contains at least one V-region, as it is otherwise not secreted (Schwarzbauer et al. 1989 PMID :2600138).

Please indicate in Figure legend 3, the nature of (SMFH2) (Line 1046)

Lines 259-260 : this section discusses the retrograde speed of actin, how are the measured speeds compare to other studies of actin retrograde flow in lamellipodia and filopodia?

Line 307: Please check formatting of references 31 & 32

Lines 384 and 385 : Please check grammar and meaning of sentence.

Line 454 : check spelling of « somitogenesis ».

<Point-by-point response>

Our answer to Reviewer #2 is colored in blue.

REVIEWERS' COMMENTS

Reviewer #2 (Remarks to the Author):

Review to Jo et al.

I would like to thank the authors for their efforts to explain the experimental results in detail and for their efforts to try even another linear RGD-peptide for their binding studies.

We appreciate the reviewers' comment so far.

My main problem to understand the presented data, was to follow the fate of the activated and ligand-bound integrins on the 12pN-RGD-tether. If these ligands are strongly bound to ligands, and are able to couple to the retrogradely moving actin cytoskeleton, we should be able to observe extensive rupture of the low-force tethers. This rupture, however, is not occurring and explanations for this effect are still missing. One solution to this problem could be that despite the ligand induced out-side in activation, the cell is not able to couple these integrins to the rearward moving actin cytoskeleton. I therefore asked the authors to put cells on substrates simultaneously presenting low and high-force RGD tethers. As nicely shown by the authors, cell shape, and frequency of tether ruptures are completely different to the conditions with individual force tethers. Cells do not fully spread on such mixtures, and show extensive rupture of the 12pN-RGD-ligands, occurring at the leading edge of the spread cell, very similar to what is shown on the low-force tethered LDV-ligands.

That spreading is not complete is an indication that the rupturing of low-force tethers is inducing a retro-control on the actin cytoskeleton, controlling cell spreading also on softer surfaces. In addition, this experiment clearly shows that 12pN-RGD-tethers can be recognized by RGD-integrin and are pulled from the substrate, when seen in the context of the leading edge of the lamellipodium, but apparently not during the initial phases of cell-substrate interactions via filopodia.

The reviewer's recapitulated the suggestion on mixed ligand condition and our updated data in the previous round of revision. In the mixed condition, the frequency and density of force transmission (RGD-12 rupture) are relatively much lower in the initial phase.

In another key experiment, the authors attach the cells to the substrate, via a non-integrin-dependent cholesterol-linker, and observed significant activation (EO conformation) of the RGD-dependent integrins, without inducing spreading, nor significant rupture of the 12-pN tether. Only when the tether force-resistance is above 33-pN, cells also begin to spread on these surfaces. As the authors correctly state, the activation state of the integrins is not coupled to the signaling/spreading activity, nor tether rupture events.

While the authors put a big emphasis on the dissection of the extracellular mechanisms, the key difference between $\alpha 4\beta 1$ and $\alpha v\beta 1$ (or $\alpha 5\beta 1$) is likely to be situated at the interface between the integrin cytoplasmic tails and the retrogradely moving actin filaments. As nicely shown, perturbation of the lamellipodial F-actin flow (by the formin inhibitor), but not myosin-activity, blocks the spreading/rupture events. This puts the finger on the integrin/F-actin link and the potential reason for the observed differences. Clearly one possible explanation of the differences between $\alpha 4\beta 1$ and $\alpha 5/\alpha v\beta 1$ integrins are the observation that the $\alpha 4$ -cytoplasmic tail is special

(several references), having notably the capacity to recruit paxillin directly. Possibly, direct paxillin recruitment to integrins could induce integrin-mediated signaling that otherwise needs to be provided by the tension-dependent recruitment of paxillin to the integrin/talin/kindlin complex. Relatively recent data, shows that paxillin is not only a signal transducer, but is directly involved in the physical maturation of the integrin/talin/actin linkage, and can when directly recruited to the integrin receptor induce focal adhesion maturation (see Ripamonti et al., 2021). The leukocyte expressed $\alpha 4\beta 1$ integrin is also special concerning its requirement for RIAM-mediated talin activation. While $\beta 2$ -integrin (LFA-1) activation requires Rac1/RIAM, RIAM is not required for $\alpha 4\beta 1$ integrin-dependent leukocyte adhesion. Although $\beta 2$ -integrins are not discussed in this manuscript, they are known for their association with filamin, which binds similar to myosin-X to integrins prior to their activation via an inside-out activation mechanisms. Filamin association with $\beta 1$ integrins outside of focal adhesions has been shown by fluorescence correlation microscopy by the Horwitz group.

The reviewer suggested that the key difference between $\alpha 4\beta 1$ and $\alpha v\beta 1$ (or $\alpha 5\beta 1$), we reported, is likely to be situated at the interface between the integrin cytoplasmic tails. We agree that the integrin tails would be involved in two different mechanotransduction and cell spreading patterns we reported. This is interesting and important topic to pursue in the future. We added the discussion on the potential role of paxillin in the previous round.

Here in this manuscript, the authors show that high-affinity integrin ligands induce conformational activation of the extracellular integrin domains, well below the 12pN-force threshold. However, it is not clear when and how this conformational change is actually transmitted to the integrin cytoplasmic tails. What is happening to integrin extracellular domain, when filamin is providing the mechanical interaction to the retrogradely moving F-actin filaments? Will the integrin dissociate from the high-affinity ligand below the 12-pN-force threshold? What happens with the integrin when the linkage to the F-actin is not reinforced by the talin/kindlin linkage? Potentially, we would expect to find the integrin pulled out of the cell membrane, even by the relatively low force 12-pN tether. Pulling integrins out of the cells has been previously observed, when migrating cells are retracting from high affinity ligands, or seen in experiments in drosophila embryos, during body wall muscle contraction especially in response to a weakening of the integrin/talin linkage. These are potential aspects the authors could address in further studies, however after several revisions, the study is now sufficiently well referenced and shows the critical experimental details and data, from which the informed reader should be able to draw similar conclusions, to the once, I just mentioned.

The suggested the potential aspects and follow up research topics. Thanks for the suggestions and careful reading of our manuscript.

Minor issues:

Line 142 : the statement that plasma fibronectin is not expressing the $\alpha 4\beta 1$ binding V-segment is not correct: Plasma fibronectin does not express the EIIIA and EIIB domains, but the fibronectin dimer contains at least one V-region, as it is otherwise not secreted (Schwarzbauer et al. 1989 PMID :2600138).

Thanks for the comment. We deleted the phrase.

Please indicate in Figure legend 3, the nature of (SMFH2) (Line 1046)

We described SMIF2 as a formin inhibitor in our main text. The description on inhibitors in Figure 3 is for their abbreviations in the figure.

Lines 259-260 : this section discusses the retrograde speed of actin, how are the measured speeds compare to other studies of actin retrograde flow in lamellipodia and filopodia?

>> Our data (10-25 nm/s) is comparable to that of U2OS cell on fibronectin (0.9 $\mu\text{m}/\text{min}$ \sim 15 nm/s) measured with the similar method (Dylan T. Burnette 2014).

Line 307: Please check formatting of references 31 & 32

>> Thanks, we corrected the superscript error.

Lines 384 and 385 : Please check grammar and meaning of sentence.

We edited as shown below.

β 1 integrin activation quantified here by the number of bound AF647-labeled 12G10 Fab (supplementary Fig. 4a) is interesting to compare to the number of ruptured TGT. As 12G10 Fab (20 nM) ~~diffuses under adherent cells and~~ binds to and dissociates from the EO state of β 1 in the time scale of 2-3 min (supplementary Fig. 11), 12G10 Fab binding after 30 min measures the number of substrate-bound, active β 1 integrins at steady state.

Line 454 : check spelling of « somitogenesis ».

>> We corrected it.